# A plastid two-pore channel essential for inter-organelle communication and growth of *Toxoplasma gondii*

Zhu-Hong Li [1], Thayer P. King[1], Lawrence Ayong[1], Beejan Asady[2], Xinjiang Cai [3], Taufiq Rahman [4], Stephen A. Vella [1], Isabelle Coppens[2], Sandip Patel [5] & Silvia N. J. Moreno [1,6 ✉]

Two-pore channels (TPCs) are a ubiquitous family of cation channels that localize to acidic organelles in animals and plants to regulate numerous $Ca^{2+}$-dependent events. Little is known about TPCs in unicellular organisms despite their ancient origins. Here, we characterize a TPC from *Toxoplasma gondii*, the causative agent of toxoplasmosis. TgTPC is a member of a novel clad of TPCs in Apicomplexa, distinct from previously identified TPCs and only present in coccidians. We show that TgTPC localizes not to acidic organelles but to the apicoplast, a non-photosynthetic plastid found in most apicomplexan parasites. Conditional silencing of TgTPC resulted in progressive loss of apicoplast integrity, severely affecting growth and the lytic cycle. Isolation of TPC *null* mutants revealed a selective role for TPCs in replication independent of apicoplast loss that required conserved residues within the pore-lining region. Using a genetically-encoded $Ca^{2+}$ indicator targeted to the apicoplast, we show that $Ca^{2+}$ signals deriving from the ER but not from the extracellular space are selectively transmitted to the lumen. Deletion of the *TgTPC gene* caused reduced apicoplast $Ca^{2+}$ uptake and membrane contact site formation between the apicoplast and the ER. Fundamental roles for TPCs in maintaining organelle integrity, inter-organelle communication and growth emerge.

[1] Center for Tropical and Emerging Global Diseases, University of Georgia, Athens, GA 30602, USA. [2] Department of Molecular Microbiology and Immunology, Johns Hopkins Bloomberg School of Public Heath, Baltimore, MD 21205, USA. [3] Division of Cardiology, Department of Medicine, David Geffen School of Medicine at UCLA, Los Angeles, CA, USA. [4] Department of Pharmacology, University of Cambridge, Tennis Court Road, Cambridge CB2 1PD, USA. [5] Department of Cell and Developmental Biology, University College London, London WC1E 6BT, UK. [6] Department of Cellular Biology, University of Georgia, Athens, GA 30602, USA. ✉email: smoreno@uga.edu

 1

Two-pore channels (TPCs) constitute a family of intracellular cation channels that localize to acidic organelles such as animal lysosomes and plant vacuoles[1,2]. They contain two repeats of a six-transmembrane pore-forming domain, and thus occupy a key intermediate position in the evolution of multi-domain voltage-gated ion channels[3]. In deuterostome animals, three genes are typically present but the gene encoding TPC3 has undergone lineage-specific loss and is a pseudogene in humans[4,5]. Mammalian TPC1 and TPC2 are the target channels for the $Ca^{2+}$ mobilizing messenger NAADP[6–9]. They regulate numerous processes including prominent roles in membrane trafficking and organelle morphology[10] likely through associated NAADP binding proteins[11]. But some TPCs may also be activated by the endosomal phosphoinositide, $PI(3,5)P_2$[12]. Recent work suggests that TPC2 can switch its ion selectivity in an agonist-dependent manner such that it forms a non-selective cation channel permeable to $Ca^{2+}$ when activated by NAADP and a highly selective $Na^+$ channel when activated by $PI(3,5)P_2$[13]. Most plants possess a single copy of TPC that is structurally similar to animal TPCs but distinguished by the presence of two EF-hands[14–16]. Arabidopsis TPC is co-regulated by $Ca^{2+}$ and voltage, is a non-selective cation channel and regulates important processes like plant growth, germination and salt-stress[1,14]. TPCs have been described in a number of unicellular organisms[3]. But little is known about these channels in protists[17].

Apicomplexan parasites include the etiologic agents of malaria (Plasmodium spp.), a life-threatening disease, and toxoplasmosis (Toxoplasma gondii), a disease that affects primarily the fetus of pregnant women and immunocompromised patients. T. gondii, as most apicomplexans, possesses a number of features found in plants including a non-photosynthetic plastid termed the apicoplast[18] and a number of plant-like enzymes, some of which localize to the plant-like vacuole (PLV, also termed vacuolar compartment or VAC)[19,20]. The apicoplast has been proposed to originate by secondary endosymbiosis of an ancestor that ingested a red alga thereby explaining the presence of four membranes[21]. The outermost membrane is derived from the host endosomal compartment[21]. The apicoplast houses pathways for fatty acid (FASII pathway) and isoprenoid (deoxy-xylulose phosphate (DOXP) pathway) synthesis, iron-sulfur cluster assembly, and a segment of the heme synthesis pathway[22]. It is therefore an essential organelle with considerable potential for tackling parasitic infection.

In this work, we identify the T. gondii TPC and show that it localizes to the apicoplast. Using reverse genetics approaches, we reveal a critical role for pore activity in apicoplast biogenesis and parasite replication. We also uncover a novel mechanism whereby TPC mediates functional and physical coupling between the apicoplast and the ER through inter-organellar $Ca^{2+}$ transfer.

## Results

**Toxoplasma possesses a novel TPC.** We cloned a full-length cDNA of a putative TPC from T. gondii (TGGT1_311080)[23] by reverse transcription-PCR. Querying of genomic sequences with this sequence (TgTPC) identified a number of homologs in other isosporoid coccidians (e.g. Neospora) and eimeriids (e.g. Eimeria) (Supplementary Table 1) but not in hemosporidians such as Plasmodium. To probe the evolutionary relationship between coccidian TPCs and other TPCs, we performed phylogenetic analysis of the newly identified TPCs with select animal and plant TPCs together with unicellular related TPCs (TPCRs)[3]. As shown in Fig. 1a, coccidian TPCs did not group with known TPCs but instead formed a distinct clade. TPC homologs were present in chromerids, close free living ancestors of apicomplexa suggesting that the absence of TPCs in hemosporidians was due to lineage-specific loss.

The open reading frame of TgTPC corresponds to a predicted protein of 1,502 amino acids with an apparent molecular weight of 160 kDa. TgTPC possesses two ion channel domains typical of other TPCs. However, the N-terminus and the linker between the channel domains are extended. The latter lacked EF-hand domains found in plant TPCs (Fig. 1b). TgTPC showed significant overall sequence similarity to other TPCs in the pore regions (Fig. 1c). However, the selectivity filter region in both domains deviated from animal and plant TPCs. Leucine in the first pore helices of both repeat domains that was shown previously to be required for channel activity of animal TPCs[4], however, was conserved (Fig. 1c, d). TgTPC represents a novel member of the TPC family.

**TgTPC localizes to the apicoplast.** To investigate the localization and function of T. gondii TPC, we introduced a triple HA epitope-tag at the 3′ terminus of the TgTPC locus and isolated TPC-3HA single cell clones (Fig. 2a). Additionally, the 5′ promoter region of these clones was replaced with a tetracycline-regulatable element[24] in the cell line termed iΔTPC-3HA (Fig. 2b) where the expression of tagged TgTPC could be controlled by anhydrotetracycline (ATc). These genetic modifications were done in the TatiΔku80 background cell line that combines regulated gene expression[24] with high efficiency of homologous recombination[25]. A number of cell lines were generated for this work and they are listed in Supplementary Table 2.

Southern blot analysis confirmed the insertion of both the tag and promoter element in clonal cell lines (Supplementary Fig. 1). Western blot analysis using anti-HA antibodies revealed a band of ~150 kDa in both lines consistent with the predicted size of TgTPC (Fig. 2c). Note that the protein levels were lower in the TgTPC-3HA line than in the iΔTPC-3HA mutant consistent with the weaker nature of the endogenous promoter. Addition of ATc to cultures of the iΔTPC-3HA mutant down-regulated the expression of TgTPC (Fig. 2c) such that at 48 h after addition of ATc, TgTPC protein was no longer detectable. Immunofluorescence analysis (IFA) revealed HA staining of a distinct organelle that also labeled with DAPI, suggesting a DNA containing organelle like the apicoplast, in both lines (Fig. 2d). Somewhat surprisingly, we found little overlap of staining with cathepsin L, or the vacuolar-$H^+$-pyrophosphatase, both markers of the PLV (Supplementary Fig. 2a, b). The PLV is an important acidic organelle where TPCs normally reside in other organisms. Instead, we found substantial co-localization with Hsp60 (Fig. 2d), a luminal marker of the apicoplast[26].

To further probe the localization of TgTPC, we performed super-resolution microscopy. In these experiments, we co-labeled the iΔTPC-3HA line with antibodies to HA and a different apicoplast marker, the acyl carrier protein (ACP)[27]. As shown in Fig. 2e, TgTPC appeared to surround ACP suggesting a non-luminal localization. Consistent with this, cryo-electron microscopy showed expression of TgTPC on peripheral apicoplast membranes (Fig. 2f, arrowheads and Supplementary Fig. 2c). Quantification of gold labeling from 25 images showed 115 marks on apicoplast membranes compared to 35 in the apicoplast lumen and 22 outside the apicoplast. In sum, these data identify TgTPC as a novel apicoplast protein.

**TgTPC is required for apicoplast integrity and function.** To investigate the physiological role of TgTPC, we down-regulated the expression of TgTPC in the iΔTPC-3HA mutant with ATc. As expected, IFA of tachyzoites cultured in the presence of ATc for 3, 4 or 7 days showed that TgTPC was not expressed (Fig. 3a, red signal). Labeling of two apicoplast markers, ACP (Fig. 3a, +3d, green) and Hsp60 (Fig. 3b, +4d, green) were still evident 3 or

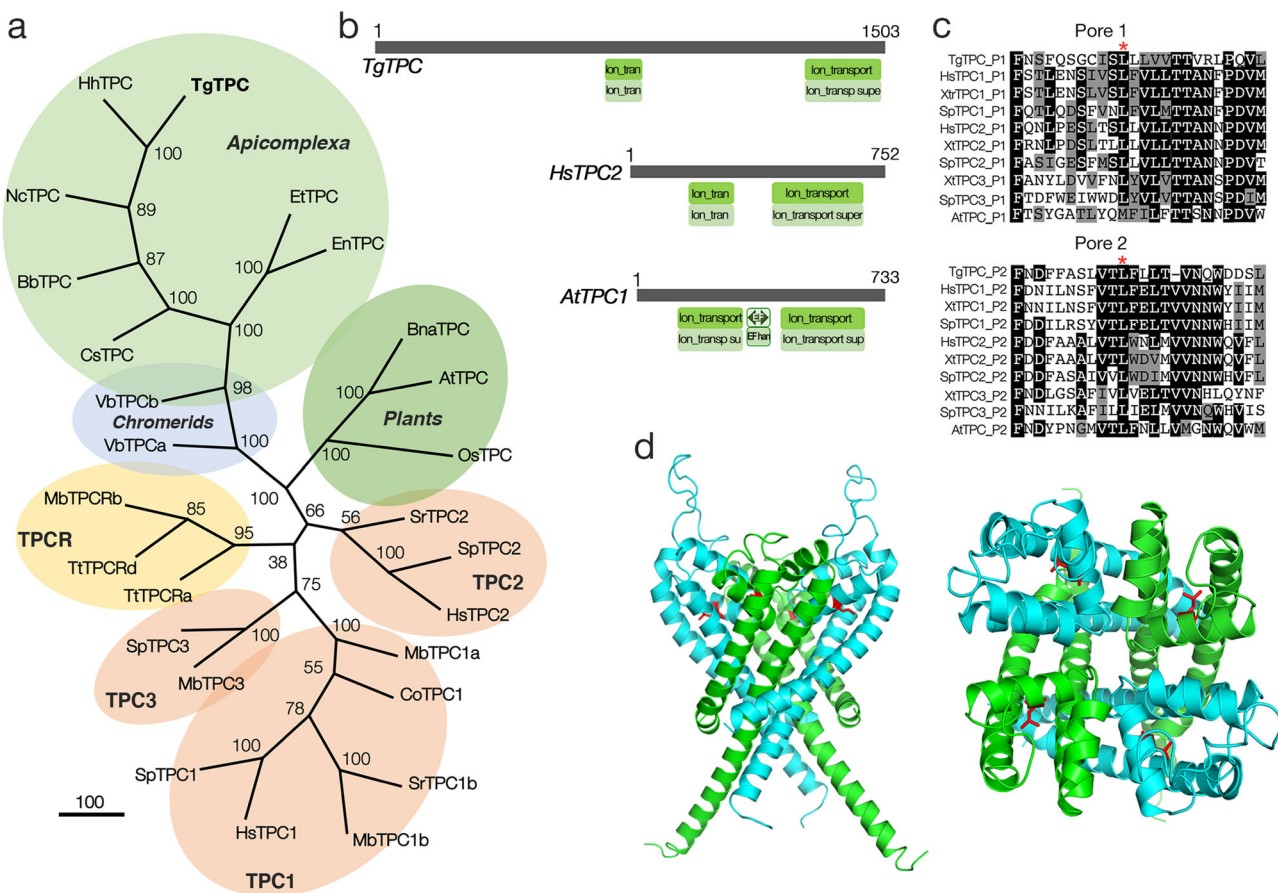

**Fig. 1 Phylogenetic placing and predicted structure of a novel TPC in *Toxoplasma*. a** Phylogenetic analysis of TgTPC. Cladogram of TPC and TPC-related (TPCR) sequences from coccidians and other select unicellular organisms, metazoans and plants. Abbreviations used: Tg, *Toxoplasma gondii*; Et, *Eimeria tenella*; En, *Eimeria necatrix*; Nc, *Neospora caninum Liverpool*; Hh, *Hammondia hammondi strain H.H.34*; Bb, *Besnoitia besnoiti*; Cs, *Cystoisospora suis*; Vb, *Vitrella brassicaformis* CCMP3155; At, *Arabidopsis thaliana*; Os, *Oryza sativa* Japonica Group; Bn, *Brassica napus*; Sr, *Salpingoeca sp. ATCC 50818*; Tt, *Thecamonas trahens;* Co: *Capsaspora owczarzaki*; Mb, *Monosiga brevicollis*; Sp, *Strongylocentrotus purpuratus*; Hs, *Homo sapiens*. Accession numbers are listed in Supplementary Table 1. **b** Predicted domain architecture of TgTPC. Schematic of TgTPC and TPCs from *Homo sapiens* (HsTPC2) and *Arabidopsis thaliana* (AtTPC) highlighting ion channel and EF-hand domains. **c** Conservation of the TgTPC pore. Multiple sequence alignment of the pore helix and selectivity filter region in the first (Pore 1) and second (Pore 2) domain of TgTPC and select animal and plant TPCs. A conserved Leucine residue (*) required for channel activity is highlighted. Abbreviations used: Tg, *Toxoplasma gondii*, Hs, *Homo sapiens*; Xt, *Xenopus tropicalis*, Sp, *Strongylocentrotus purpuratus*, At, *Arabidopsis thaliana*. **d** Structural model of the TgTPC1. Side (left) and luminal view of the TgTPC pore highlighting conserved leucine residues (red) in domain I (green) and domain II (blue).

4 days with ATc, respectively. Interestingly, the apicoplast αhsp60 signal (usually stronger than αACP) was no longer detectable after culturing *iΔTPC-3HA* with ATc for 7 days (Fig. 3a, +7d). Quantitative and morphological analysis of intracellular parasites for the presence of apicoplast is shown in Supplementary Fig. 3a, b. This analysis showed a decrease in the number of apicoplasts labeled with αhsp60 compared to the parental line, in the presence of ATc, beginning at 3 days (+3d). Almost all intracellular parasites lost labeling with αhsp60 at 7 days + ATc (Supplementary Fig. 3a, +7d, middle panel). In contrast, conditional knock down of hydroxyacyl-CoA dehydratase (DEH)[28], an essential cytosolic enzyme did not affect apicoplast integrity (Supplementary Fig. 3a, lower panel).

To characterize further the apicoplast phenotype, we examined tachyzoites following 4 days of ATc treatment. As shown in Fig. 3b apicoplast labeling became diffuse and additional labeling of Hsp60 in the residual body of intracellular parasites was observed (Fig. 3b, *arrow*). The presence of apicoplast markers in the residual body indicates a defect in segregation of the organelle during cell division[29,30]. Transmission electron microscopy of the

mutant *iΔTPC-3HA* cultured with ATc for four days showed that the apicoplast membranes were present but the organelle appeared vacuolated (Fig. 3c and Supplementary Fig. 3c). To examine the consequences of downregulating TgTPC on apicoplast function, we examined lipoylation of the pyruvate dehydrogenase subunit E2 (PDH-E2). This enzyme localizes to the apicoplast and is modified by lipoic acid produced by the FASII pathway. Western blot analysis with anti-lipoylated PDH-E2 antibody[31] (Fig. 3d) revealed that the lipoylation was reduced at days 3 and 5 compared to controls in the *iΔTPC-3HA (+ATc)* and was absent after 7 days with ATc (Fig. 3d). This phenotype could result from defective lipoylation or from reduced levels of the apicoplast protein resulting from defective segregation of the organelle.

We next measured growth and general fitness of TgTPC-depleted parasites. Growth was assessed in two ways. In the first approach, we performed plaque assays, in which the parasite engages in repetitive cycles of invasion, replication, and egress causing host cell lysis and formation of plaques observed as white spots by staining with crystal violet. As shown in Fig. 4a, ATc had a substantial effect on the growth of the *iΔTPC-3HA* mutant

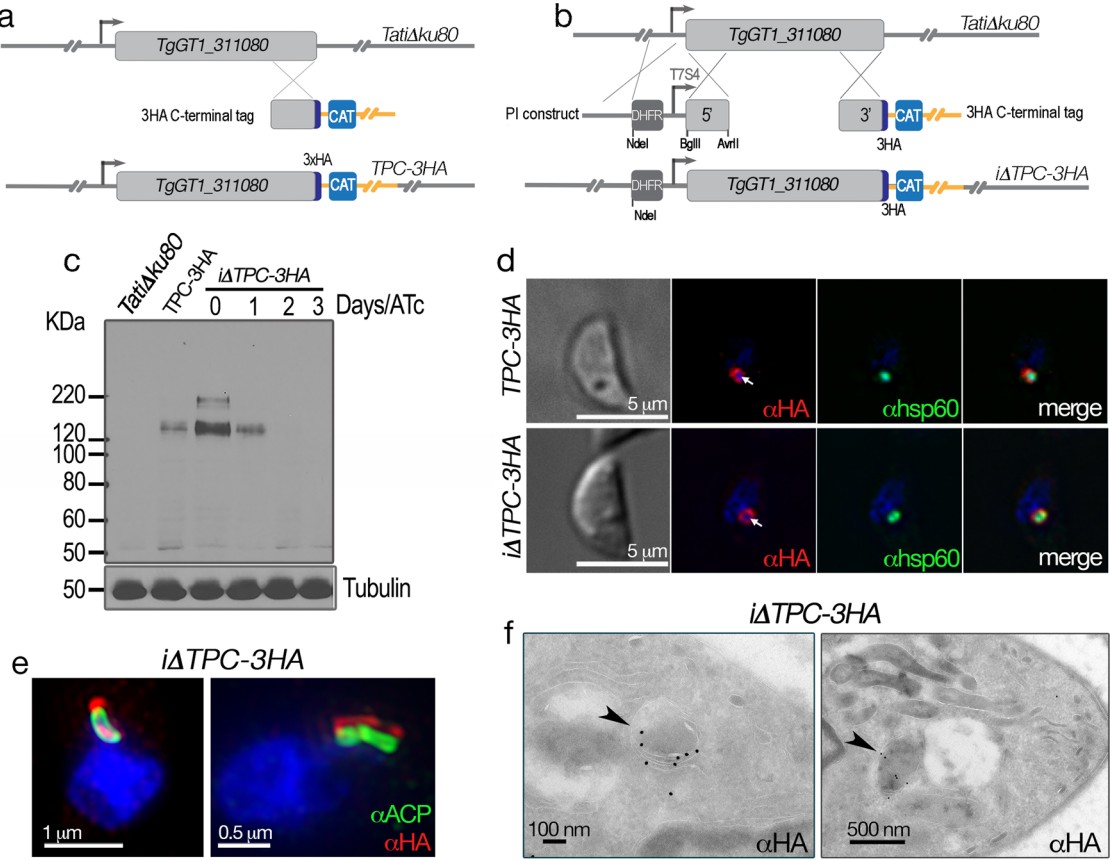

**Fig. 2 The *Toxoplasma* TPC localizes to the apicoplast. a** Cartoon showing in situ 3′ tagging of the *TgTPC* gene. **b** Diagram showing the genetic modifications of the *TgTPC* gene by insertion of a regulatable promoter at the 5′ end of its coding sequence and the addition of the 3xHA at the 3′ end. DHFR, dihydrofolate reductase gene for pyrimethamine selection. CAT, chloramphenicol acetyl transferase for chloramphenicol selection. **c** Western blot analysis with αHA antibody (Rat antibody at 1:200) showing that ATc downregulates the expression of TgTPC to undetectable levels at 2 days. αTubulin antibody (1:5,000) was used to control loading. $N = 3$. **d** IFAs of cells expressing *TgTPC-3HA*. αHA was used at 1:25 dilution (*red signal*). The apicoplast marker was Hsp60 and the antibody αhsp60 was used at 1:1,000. The HA signal co-localizes with the apicoplast marker as well as the DAPI signal characteristic of the apicoplast. The white arrow points to the DAPI signal from the apicoplast DNA which is surrounded by the red signal. $N = 3$. **e** IFAs of the *iΔTPC-3HA* mutant showing co-localization of the HA with the apicoplast marker. These images were obtained with a super-resolution microscope. The antibody used was αHA at a 1:25 (*red signal*). The apicoplast marker was ACP and the αACP antibody was used at a 1:200 dilution (*green signal*). $N = 2$. **f** Immunoelectron microscopy with a αHA antibody showing TgTPC localization at the apicoplast identified by the characteristic 4 membranes (arrowheads). $N = 2$.

(+ATc) but not on the parental cell line (*TatiΔku80*). Only very small plaques were observed upon downregulation of *TgTPC* expression (Fig. 4a). In the second approach, we expressed a cytosolic marker (tdTomato) in parental and mutant (*iΔTPC-3HA*) lines and used its fluorescence as a proxy for the growth of clonal cells[27]. Downregulation of the expression of *TgTPC* with ATc substantially decreased parasite growth (Fig. 4b). The growth of these mutants was reduced to almost 15 % after 7 days with ATc. Next, we evaluated intracellular replication by quantifying the number of parasites per Parasitophorous Vacuole (PV). As shown in Fig. 4c, there was a significant reduction in the number of PVs harboring 8 or 16 parasites when the cells were pre-treated with ATc for 4 days with a more noticeable difference at day 7 (Fig. 4c, d). We also examined invasion and egress, two critical steps of the lytic cycle. Egress, stimulated with ionomycin, was significantly prolonged when the *iΔTPC-3HA* cells were cultured with ATc for 24 h following 4 days pre-incubation (total 5 days) (Fig. 4e). Invasion, evaluated by a red-green assay, was normal at day 4 but markedly reduced after 7 days with ATc (Fig. 4f). All steps of the lytic cycle therefore were affected following extended TPC downregulation and reminiscent of the delayed death phenotype previously observed for apicoplast enzyme inhibitors

and genetic disruption of genes required for apicoplast biogenesis or metabolism[30].

Collectively, these data show that TgTPC is essential for maintenance of apicoplast integrity and parasite growth.

**Isolation of *TPC* null mutants**. Because the apicoplast is an essential organelle[32], we sought ways of dissecting physiological roles for TgTPC independent of the apicoplast-related phenotypic changes induced by downregulating TgTPC. To this end, we manually expanded the conditional mutants from the small plaques for 6–8 weeks until the parasites were able to complete the lytic cycle on their own (see scheme in Supplementary Fig. 4 and detailed protocol in Supplementary information). This strategy was followed with two cell lines *iΔTPC* and *iΔTPC-3HA*, which were grown with ATc and the cell lines generated were termed *iΔTPC-TR and iΔTPC-3HA-TR* for Tetracyclin Resistant because after these passages, the small plaques formed by the mutants were not further reduced by ATc. The signal of the apicoplast appeared to be absent in the *iΔTPC-3HA-TR* line (Supplementary Fig. 4b). Because the genomic locus of the *TgTPC* gene was still intact in both mutants, we deleted approximately ~5 kb of the *TgTPC* gene of the *iΔTPC-TR* line by homologous recombination

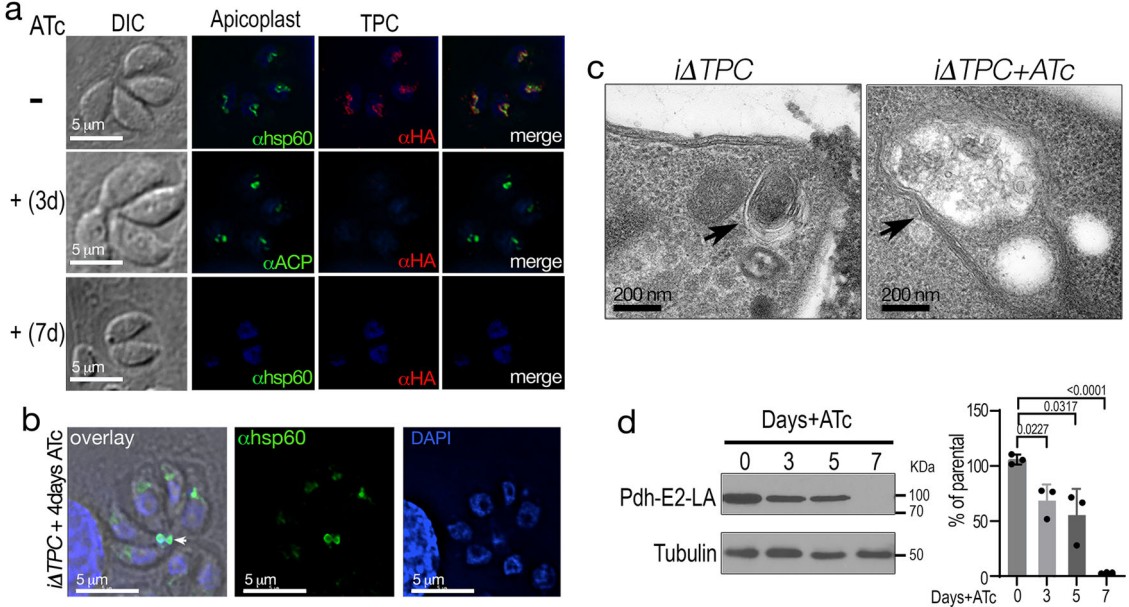

**Fig. 3 Downregulation of the *TgTPC* impacts integrity and function of the apicoplast. a** IFA with αHA showing that TPC is not expressed after 3-days of culture with ATc. The signal corresponding to the apicoplast is still present as detected with αACP. However, after 7 days with ATc the apicoplast marker Hsp60 is no longer detectable. $N = 3$. **b** labeling of the apicoplast components at the residual body 4 days after ATc treatment. Labeling of the apicoplast was done with αhsp60 (1:1,000). *Arrow* shows the αhsp60 and DAPI signals at the residual body. $N = 3$. **c** Routine electron microscopy of the *iΔTPC* grown in the presence of ATc, showing that the apicoplast becomes vacuolated. $N = 2$. **d** Lipoylation of PDH-E2 was detected by western blot analysis with an anti-lipoylated-PDH-E2 antibody (clone3H-2H4 1:1,500 dilution). *iΔTPC* cells treated with ATc showed decreased lipoylation of PDH-E2. $N = 3$. Student's *t*-test was used: 3 versus 0 days, $p = 0.0227$. 5 versus 0 days, $p = 0.0317$. 7 versus 0 days, $p < 0.0001$. Data are presented as mean ± SD. $p$ value: unpaired two tailed t test performed in all comparisons.

with an engineered cosmid and isolated clones by limiting dilution (Fig. 5a). Southern blots (Supplementary Fig. 5a) confirmed deletion of the *TgTPC* gene. These clones, could be stably maintained in culture but they grew at a very low rate. The *ΔTPC-a* clone was cultured for several months (6-12 months) and, interestingly, apicoplast labeling with ACP was recovered (Fig. 5b). This was reproduced with an independent mutant, *ΔTgTPC-c* (Supplementary Fig. 5b). Apicoplasts were visible by EM in the *ΔTPC-a* mutant (Fig. 5c). In accord, the *ΔTPC-a* mutant showed apicoplast specific lipoylation activity (Fig. 5d).

**Adaptive changes in TPC null mutants are associated with changes in apicoplast DNA content**. We considered a potential genomic compensatory change resulting in the recovery of apicoplast function in the *ΔTPC-a* mutant. To investigate this, we performed whole-genome sequencing of the mutant and mapped it to the reference strain TgGT1 (ToxoDB)[33]. Two addtional clones (*ΔTPC-b* and *ΔTPC-c-5*) plus the original tetracycline resistant (*iΔTPC-TR*) and the *iΔTPC* were also sequenced for comparison (Supplementary Table 3). The analyzed sequences did not reveal any notable change in their genome except for a few indels within intergenic regions.

We then investigated if the apicoplast DNA content in the *ΔTPC* mutant played a role in its adaptation. We had observed that the *ΔTPC* clones contained varying levels of apicoplast DNA (Supplementary Fig. 5c, d) and the *ΔTPC-a* mutant showed high apicoplast DNA content after being in culture for several months (Supplementary Fig. 5d). We used the sequencing data and estimated the copy numbers of apicoplast DNA per cell by calculating the ratio of apicoplast DNA coverage to the haploid nuclear genome (assuming that apicoplast and nuclear DNA were amplified and sequenced at equal rate) (Supplementary Fig. 6a). This result confirmed the variable amounts of apicoplast DNA previously seen in the mutants. The TR cell line which showed

almost no apicoplast labeling by IFA, had the lowest apicoplast DNA content (only ~2–3 copies per cell). The two clones of *ΔTPC*, from which genomic DNA was extracted very early after subcloning, had slightly higher apicoplast DNA than the TR line. A subclone of *ΔTPC-c* named *ΔTPC-c-5* that was cultured longer showed the highest apicoplast DNA content.

To further evaluate temporal changes in apicoplast DNA, we re-evaluted the apicoplast DNA of the *ΔTPC-a* mutant 1 month after subcloning (*ΔTPC-a-1m,*) and compared it with the same clone cultured in vitro for one year (*ΔTPC-a-12m*). The *ΔTPC-a-12m* cells possessed apicoplast DNA at the same level as *iΔTPC* parasites while the *ΔTPC-a-1m* mutant showed much lower levels of apicoplast DNA (Supplementary Fig. 6b).

We also examined temporal changes of apicoplast DNA in *iΔTPC-3HA-TR* cells when cultured with ATc (Supplementary Fig. 6c). We observed a decrease of apicoplast DNA after 4 days. When the *iΔTPC-3HA-TR* mutant was grown for 42 days with ATc, the apicoplast DNA was lowest (note at this stage the IFA did not show strong apicoplast labeling, Supplementary Fig. 4b) but it increased after 3 months of culture with ATc (Supplementary Fig. 6c). At this point, we observed brightly labeled apicoplasts in a proportion of the *iΔTPC-3HA-TR* mutant (Supplementary Fig. 6d). All subclones of the *iΔTPC-3HA-TR* cell line (we tested 12) showed some bright IFA apicoplast signal in a portion of the cells after 3-4 months in culture (we show clone 1 and 2 in Supplementary Fig. 6d). *ΔTPC-c-5* with the highest apicoplast DNA showed the most consistent apicoplast labeling.

Taken together, these data show that the recuperation of the apicoplast DNA positively impacted the growth of TPC-depleted cells.

**TgTPC regulates replication independent of apicoplast biogenesis**. We next analyzed the phenotype of the *ΔTPC-a* null

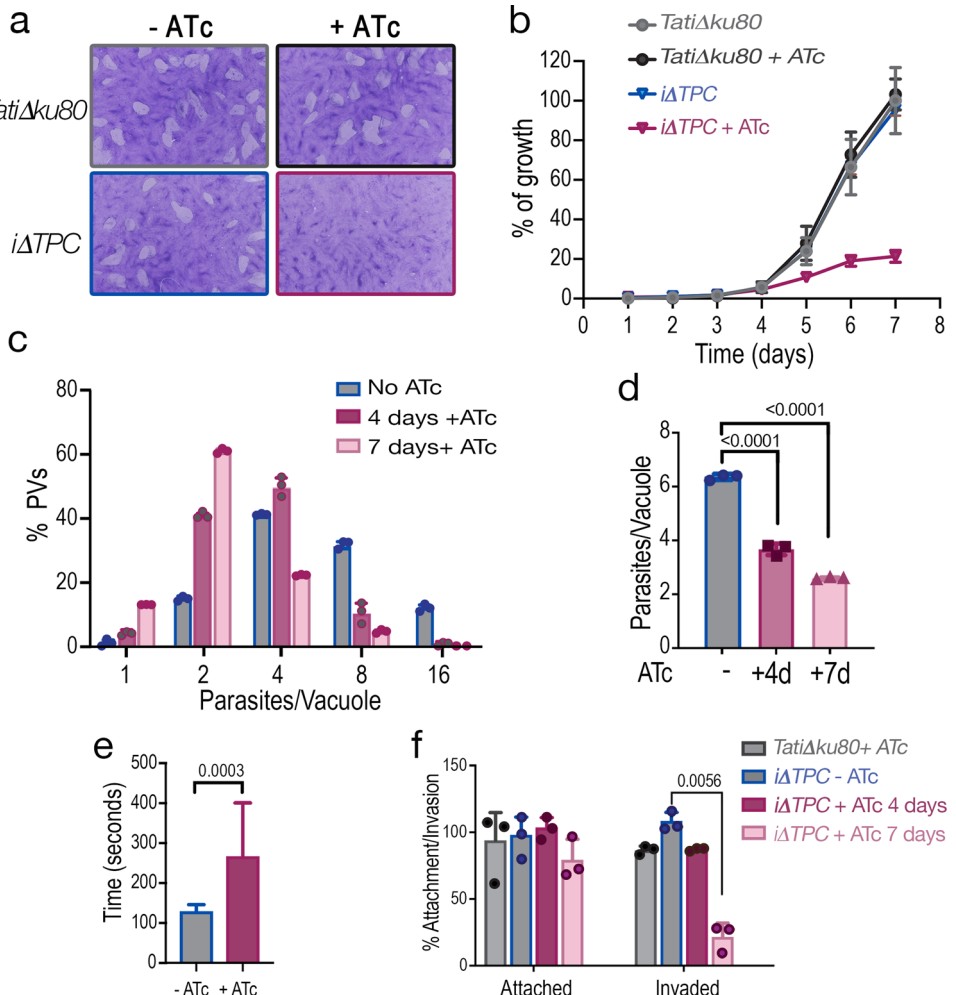

**Fig. 4 Downregulation of the *TgTPC* disrupts every step of the *Toxoplasma* lytic cycle. a** Plaque assays of the *iΔTPC* parasites cultured in the presence of 0.5 μg/ml Anhydrotetracycline (+ATc). Confluent fibroblast cells grown in 6 well plates were infected with 150 tachyzoites of the indicated cell lines for 8 days in medium ± ATc. **b** Growth of *TatiΔku80*, and *iΔTPC* in fibroblasts. All parasite lines express tdTomato (a red fluorescent protein) and were isolated as described in the Methods section. 1000 tdTomato expressing cells were used to infect each well of a 96 well plate. A standard curve for each cell line was developed for fluorescence vs. number of parasites. Growth of parental lines is shown as control: *TatiΔku80* without (*gray*) and +ATc (*dark gray*). Growth of *iΔTPC* is shown without (*blue*) and +ATc (*magenta*). N = 3. **c** Growth kinetics evaluated by counting number of parasites per parasitophorous vacuole (PVs). Percentage of PVs with 1, 2, 4, 8, and 16 parasites for each cell line at 24 h post-infection was plotted. N = 3. **d** Average number of parasites/PV 24 h post-infection at 4 days and 7 days of ATc treatment of the *iΔTPC* cells. N = 3. p < 0.0001. Error bars represent the standard deviation (SD) of three biological replicates. **e** Parasites treated with ATc for 5 days showed delayed egress when stimulated with ionomycin. 50,000 tdTomato-expressing parasites of *iΔTPC* parasites without ATc, or treated with 0.5 μg/ml ATc for 4 days were used to infect hTERT cells and allowed to grow for 24 h. n = 16, N = 4. Error bars represent the standard deviation (SD) of 16 PVs. p = 0.0003, Student's t test. **f** Invasion assays of *iΔTPC* cells treated with ATc for 4 and 7 days. Attached and invaded parasites from each cell line were normalized to the invasion of the *TatiΔku80* cells without ATc treatment (100%). N = 3, p = 0.0056, Student's t test. Data are presented as mean ± SD for (**b–f**). p value: unpaired two tailed t test performed in all comparisons.

mutant. As shown in Fig. 5e, f, growth of the *ΔTPC-a* mutant was severely retarded similar to the conditional mutant, *iΔTPC-3HA* (+*ATc*) (Fig. 4b). To determine specificity, we took two approaches. In the first approach, we complemented the *ΔTPC-a* cells with the complete cDNA of the *TgTPC* gene to generate the *ΔTPC-a-TPC* complemented mutant. The presence of TgTPC cDNA in the *ΔTPC-a-TPC* cells was confirmed by PCR (Supplementary Fig. 7). The *ΔTPC-a-TPC* clone was fully rescued and grew at a similar rate to the wild type clone (Fig. 5e, f, *green*). In the second approach, we complemented the *ΔTPC-a* cells with a conditional copy of TgTPC (Supplementary Fig. 8a, b). The resulting *iΔTPC-iTPC* mutant also showed full recovery of growth (Supplementary Fig. 8c, d, *green*). Importantly, culturing these cells with ATc to acutely down regulate TgTPC resulted in the same delayed death phenotype as the original *iΔTPC* mutant.

Thus, the addition of ATc blocked expression of the exogenous copy of TgTPC in the *ΔTPC-a-iTPC* mutant, and inhibited plaque formation (Supplementary Fig. 8c, +ATc) and growth (Supplementary Fig. 8d, *gold*). This result indicates that the adaptation of the *ΔTPC-a* mutant (and the other genome sequenced clones) most likely did not result from a compensatory mutation in its nuclear genome as this would have countered the deleterious effect of downregulating the exogenous TgTPC. However, we can not discard the possibility of other compensatory effects that could account for the differences between the phenotypes in the "adapted" *ΔTPC-a* mutant and the *iΔTPC + ATc* mutant.

Further analysis showed that host invasion (Supplementary Fig. 9a) and egress (Supplementary Fig. 9b) were not affected in the *ΔTPC-a* null mutants. This was in contrast to the significant reduction in the conditional mutant *iΔTPC + ATc* (Fig. 4c–f).

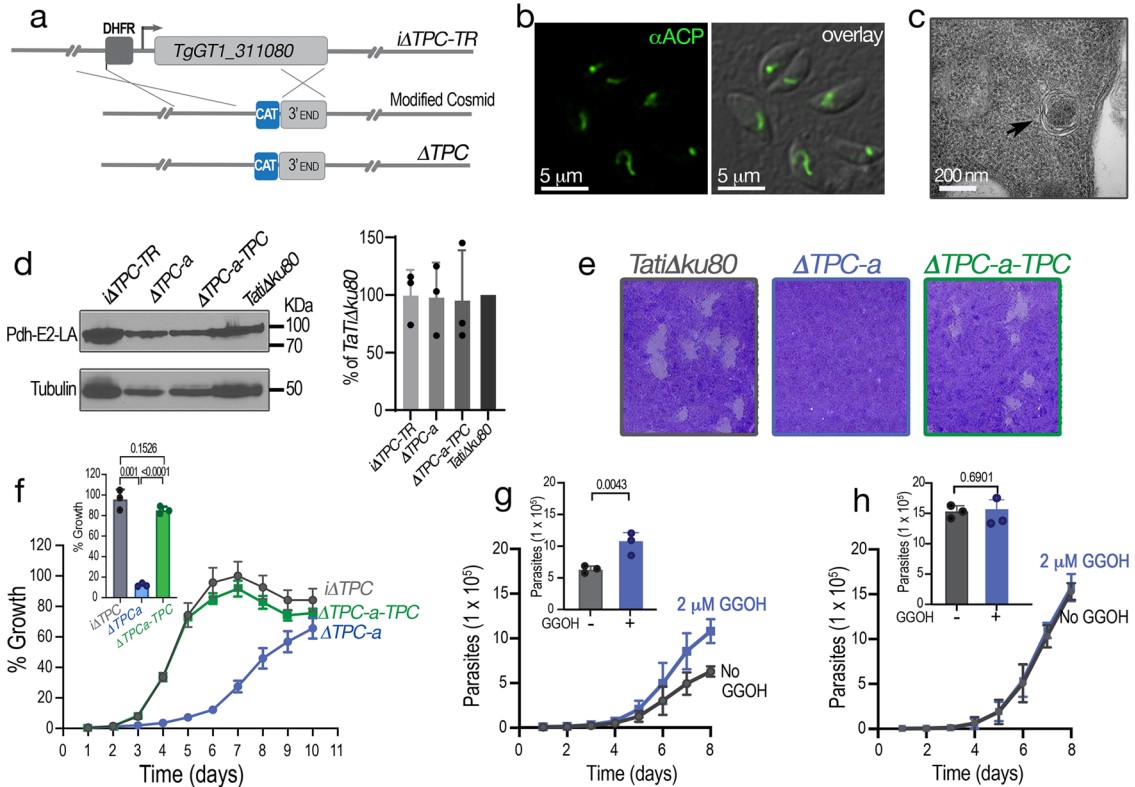

**Fig. 5 TgTPC regulates replication independent of apicoplast biogenesis. a** The *TgTPC* gene was deleted in the *iΔTPC-TR* parasites by using a cosmid strategy. The knockout cosmid construct was engineered in *E. coli* by replacing the 5′ terminus of the *TPC* gene with a chloramphenicol selection cassette. The construct containing the CAT selection cassette was transfected into the *iΔTPC-TR* parasites to replace the DHFR cassette plus 5 kb of the TPC gene by homologous recombination, generating TPC *null* mutant. **b** *ΔTPC-a* cultured for 12 months show apicoplast labeling by IFA with the marker *αACP* (1:200). N = 3. **c** Transmission electron microscopy of *ΔTPC-a* showing the characteristic apicoplast with four membranes. N = 2. **d** Lipoylation of PDH-E2 detected by western blot with an anti-lipoylated-PDH-E2 antibody (clone3H-2H4 1:1,500 dilution). N = 3, mean ± SD. **e** Plaque assays showing growth of control cells, *TatiΔku80*, compared with *ΔTPC-a* and *ΔTPC-a-TPC*. Confluent fibroblast cells grown in 6 well plates were infected with 150 tachyzoites of the indicated cell lines for 8 days. N = 3. **f** Growth of fluorescent clones of *iΔTPC, ΔTPC* and *ΔTPC-a-TPC* cells in fibroblasts in 96-well plates at 4000 parasites/ well. The % of growth of each indicated cell line was normalized to growth of *iΔTPC* (no ATc) at day 7 (considered 100%). Inset shows the quantification of three independent experiments at day 6. Error bars represent the standard deviation (SD) of the three experiments analyzed. *ΔTPC-a* versus *iΔTPC*, p = 0.0001; *ΔTPC-a* versus *ΔTPC-a-TPC*, p < 0.0001; *ΔTPC-a-TPC* versus *iΔTPC*, p = 0.1526. N = 3. Data are presented as mean ± SD. **g** Partial rescue of *ΔTPC* mutants by Geranylgeraniol (GGOH). 4,000 tdTomato expressing *ΔTPC-a cells* were used to infect each well of a 96 well plate in the presence or absence of 2 μM of GGOH. Growth was monitored daily for 8 days. N = 3. Inset shows the quantification of three independent experiments at day 8. p = 0.0043. Data is presented as mean ± SD. **h** same protocol as in **g** with 200 RH parasites. Growth was monitored daily for 8 days. Inset shows cell numbers on day 8 from 3 independent experiments. N = 3, p = 0.6901. Data are presented as mean ± SD. p value: unpaired two tailed t test performed in all comparisons.

Moreover, motility was not significantly affected upon TPC knockout in contrast to the conditional mutants (Supplementary Fig. 9c, d). These differences indicate that the slow growth phenotype of the *ΔTPC-a* mutant may result from a replication defect due to deficient production of essential metabolites like isoprenoids, which are synthesized by apicoplast enzymes[34]. These enzymes may not be working optimally because of the silenced TgTPC. In this regard, culturing the *ΔTPC-a* cells in the presence of geranylgeraniol (GGOH), an essential isoprenoid metabolite, partially rescued growth (Fig. 5g). Interestingly, GGOH had no effect on the growth of the parental cell line RH (Fig. 5h). The product of the apicoplast isoprenoid pathway, isopentenyl diphosphate (IPP) could not be tested because it does not permeate through parasite membranes[34].

Collectively, these data identify a specific role for TgTPC in replication and suggest that the effects of TPC depletion on other aspects of the lytic cycle (invasion, egress or motility) are a secondary consequence of apicoplast functional disruption.

**Regulation of replication by TgTPC requires an intact pore.** To gain further mechanistic insight into how TgTPC regulates replication, we complemented the *ΔTPC-a* mutants with regulatable copies of the TgTPC tagged with a cMyc epitope at their C-termini. We used both wild type TgTPC and mutant TgTPCs in which conserved Leu residues in either the first or second pore domain where substituted for Pro (Fig. 1c). These parasites were termed *ΔTPC-a-iTPC*[*1] and *ΔTPC-a-iTPC*[*2], respectively. Figure 6a shows IFAs of the complemented parasites using an anti-cMyc antibody. *TgTPC* expression was readily detectable in all three lines but not in the parental TPC null mutant line. Counter staining with an apicoplast marker showed that wild type and mutant TPCs localized to the apicoplast. The western blots in Supplementary Fig. 8b confirmed expression of the TgTPC at similar levels in the three lines.

Plaque analyses showed that TPC-null parasites complemented with wild type *TgTPC* formed normal plaques (Fig. 6b *middle panels, ΔTPC-a-iTPC*). This rescue provides further evidence that TgTPC is required for growth (Fig. 6c). In stark contrast, complementation of TPC-null cells with the mutated versions of

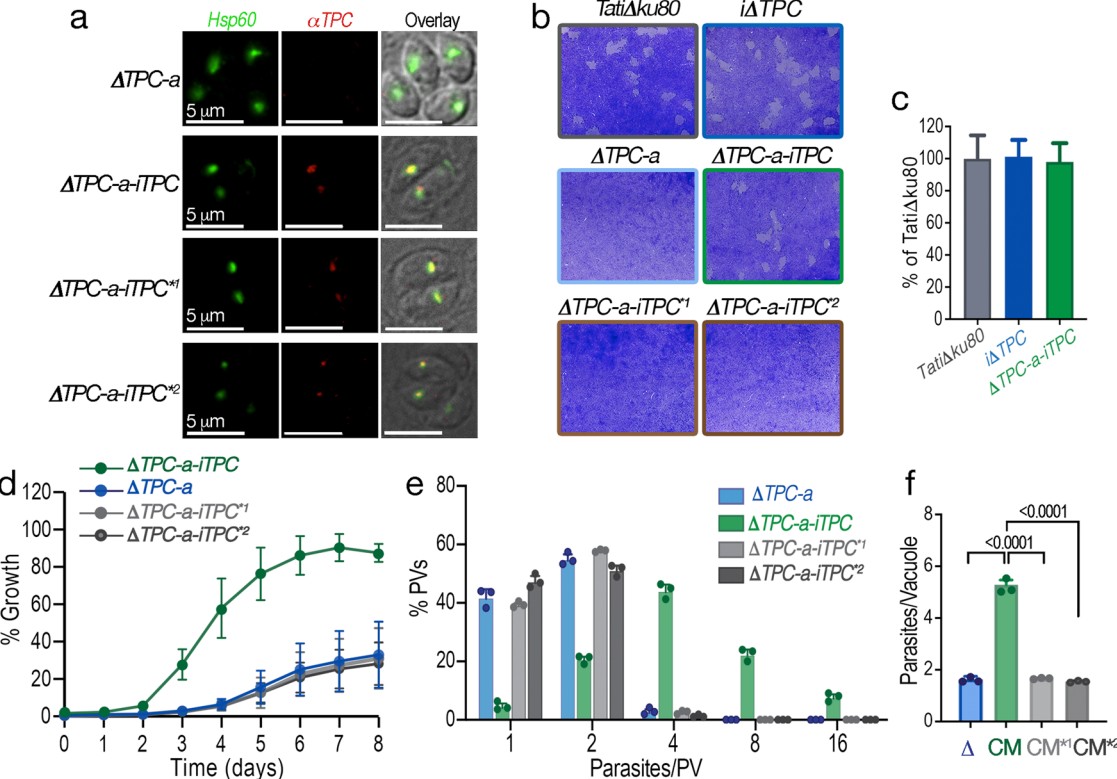

**Fig. 6 Regulation of replication by TgTPC requires an intact pore. a** IFAs showing that the complementation with the *TgTPC* cDNA (cloned in the pDT7S4my3 plasmid) generates a cell line that expresses TgTPC in the apicoplast as detected with an αMyc antibody (1:500) (*red signal*). $N = 3$. **b** Plaque assays of the parental line, *TatiΔku80, iΔTPC (no ATc), ΔTPC-a, ΔTPC-a-iTPC, ΔTPC-a-iTPC*[*1] and *ΔTPC-a-iTPC*[*2]. Confluent fibroblast cells grown in 6 well plates were infected with 150 tachyzoites of the indicated cell lines for 8 days. $N = 3$. **c** Quantification of plaque sizes generated by *TatiΔku80, iΔTPC* and *ΔTPC-a-iTPC* cell lines with ImageJ, 24 plaques examined from 3 independent experiments. Error bars represent standard deviation (SD) of the 24 plaques analyzed. No significant difference between *TatiΔku80* and *ΔTPC-a-iTPC* mutants. $p = 0.6019$, $n = 24$, $N = 3$. **d** Growth of *ΔTPC-a, ΔTPC-a-iTPC, ΔTPC-a-iTPC*[*1] and *ΔTPC-a-iTPC*[*2] cell lines in fibroblasts. All parasite lines express tdTomato and were isolated as described in Methods. 1,000 tdTomato expressing cells were used to infect each well of a 96 well plate. The % of growth of each cell line was normalized to *TatiΔku80* cells on day 8 (considered 100%). $N = 3$. **e** Growth kinetics evaluated by counting the parasites per PV 24 h post-infection. Percentage of PVs with 1, 2, 4, 8, and 16 parasites after 24 h replication of each cell line was plotted. $N = 3$. **f** Number of parasites per vacuole, 24 h after the initial infection. Comparison between *ΔTPC-a, ΔTPC-a-iTPC,* and *ΔTPC-a- iTPC*[*1] and *ΔTPC-a-iTPC*[*2] cell lines. $N = 3$. $p < 0.0001$. Data for (**c–f**) are presented as mean ± SD. *p* value: unpaired two tailed t test performed in all comparisons.

the *TgTPC* gene failed to restore plaques (Fig. 6b, *bottom panels*) indicating a requirement for conserved residues within the pore for TPC functionality. Further growth analysis of mutants transfected with the tdTomato gene also showed that the pore mutant of the TgTPC gene was unable to rescue the growth defect of the *ΔTPC-a* mutants (Fig. 6d).

To analyze further the replication defect of the *ΔTPC-a* mutant, we quantified PVs 24 h post-infection (Fig. 6e, f). Most PVs from *ΔTPC-a* parasites contained 2 parasites indicative of a replication defect (Fig. 6f, *blue bars*). In contrast, *ΔTPC-a-iTPC* parasites formed PVs containing on average ~4–8 parasites with the expected distribution indicative of a normal replication cycle (Fig. 6f, *green bars*). As with the plaque and growth kinetic analyses, complementation of *ΔTPC-a* cells with mutant TPCs failed to rescue the replication defect (Fig. 6e, f, *light and dark grey bars*).

Collectively, these data are consistent with TgTPC directly impacting *T. gondii* replication and highlight the importance of the pore domain for its function.

**TgTPC mediates selective $Ca^{2+}$ exchange between the endoplasmic reticulum and the apicoplast.** TPCs function as $Ca^{2+}$ release channels in both animal and plant cells. We therefore examined the impact of TgTPC on $Ca^{2+}$ dynamics. To directly

monitor $Ca^{2+}$ in the apicoplast lumen, we developed a novel HA-tagged reporter comprising the apicoplast targeting signal of ferredoxin oxido-reductase (FNR)[35] and the genetically-encoded $Ca^{2+}$ indicator GCaMP6f[36]. The construct was expressed in both control (RH, parental cell line of *TatiΔku80*) and *ΔTPC-a* mutant parasites and the resulting clonal cell lines generated termed *RH-FNR-GCaMP6-HA* and *ΔTPC-FNR-GCaMP6-HA*, respectively. IFA of GCaMP6-HA showed co-localization with the apicoplast marker, Hsp60 in both lines (Fig. 7a). Western blot analysis of total lysates revealed equivalent expression (Supplementary Fig. 10a). The reporter also responded similarly to the addition of the $Ca^{2+}$ chelator EGTA and $Ca^{2+}$ upon permeabilization of cell suspensions with Triton X-100 (Supplementary Fig. 10b-c).

We loaded the *RH-FNR-GCaMP6-HA* and *ΔTPC-FNR-GCaMP6-HA* tachyzoites with the chemical $Ca^{2+}$ indicator Fura-2 to simultaneously measure cytosolic (Fig. 7b–d) and apicoplast (Fig. 7e–g) $Ca^{2+}$ levels in live cells. The addition of thapsigargin (Thap), which blocks the endoplasmic reticulum SERCA pump, caused an increase in cytosolic $Ca^{2+}$ in control cells due to leak of $Ca^{2+}$ from the ER (Fig. 7b and c, *dark gray traces*). The addition of $Ca^{2+}$ after Thap (Fig. 7b) or prior to Thap (Fig. 7c) also led to an increase in cytosolic $Ca^{2+}$ due to influx from the extracellular milieu. Interestingly, when measuring apicoplast GCaMP6 fluorescence (Fig. 7e, f, *dark gray traces*

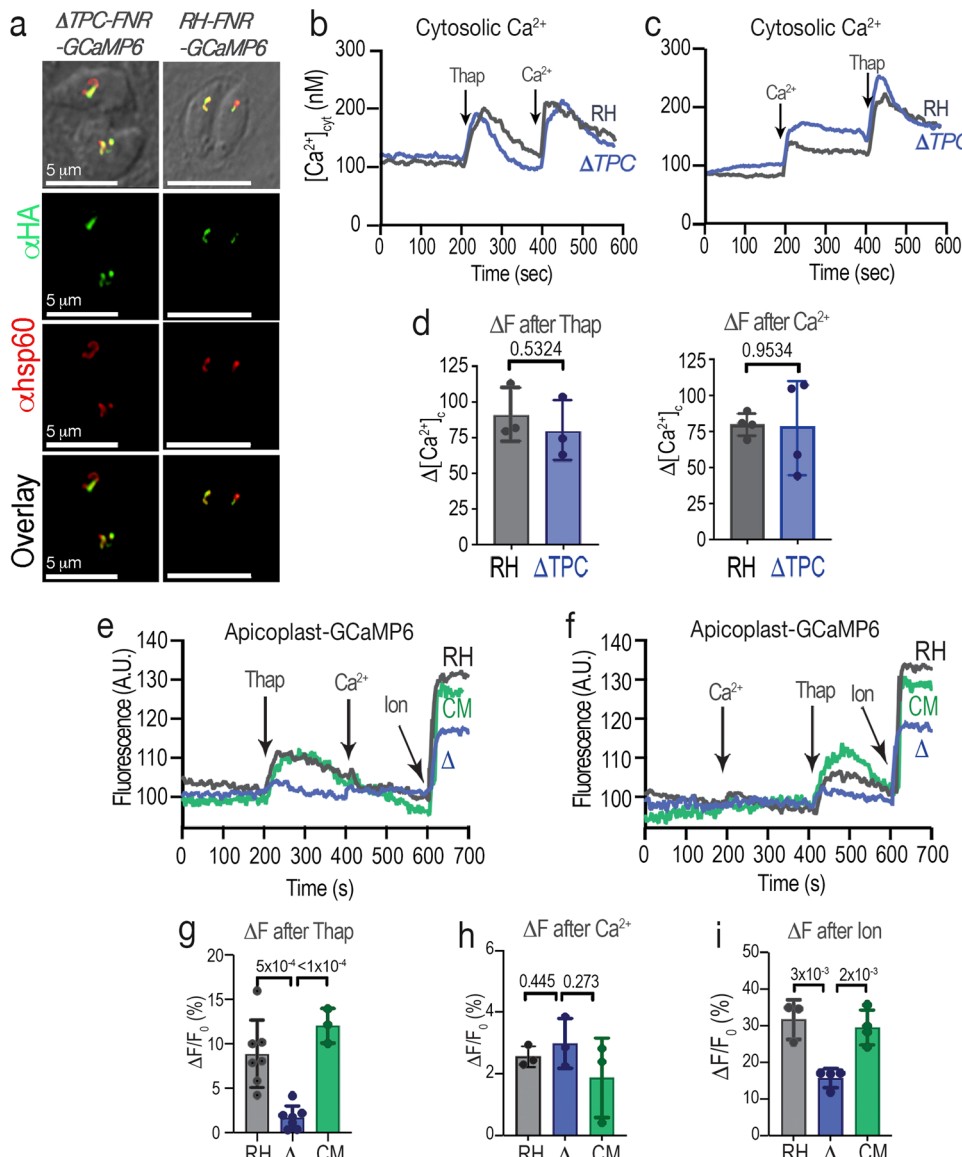

**Fig. 7 TgTPC mediates selective Ca²⁺ exchange between the endoplasmic reticulum and the apicoplast. a** IFAs of *ΔTPC-FNR-GCaMP6* and *RH-FNR-GCaMP6* parasites showing the localization of the Ca²⁺ indicator with αHA (*green*) and its co-localization with the apicoplast marker Hsp60 (*red*). $N = 3$. **b** cytoplasmic Ca²⁺ responses of Fura2-AM loaded parasites. The experiment shows the comparison between RH (*gray traces*) and *ΔTPC* (*blue traces*). Thapsigargin (Thap) and Ca²⁺ were added at the times indicate. $N = 3$. **c** similar experiment to the one in b, with the additions reversed. $N = 4$. **d** quantification of the changes in fluorescence (ΔF slopes) after the addition of 1 μM thapsigargin (ΔF after Thap) or 1.8 mM Ca²⁺ (ΔF after Ca²⁺). The response to these two additions is not significantly different between both cell lines. ΔF after Thap ($p = 0.5324$, $N = 3$); ΔF after Ca²⁺ ($p = 0.9534$, $N = 4$). Data are presented as mean ± SD. **e, f** apicoplast GCaMP6f fluorescence changes of *RH-FNR-GCaMP6* (parental, RH), *ΔTPC-FNR-GCaMP6* (Δ) and *ΔTPC-TPC-FNR-GCaMP6* (complemented, CM) cells in response to the addition of 1 μM Thap, 1.8 mM Ca²⁺ or 1 μM ionomycin (Ion). **g** Quantification of the ΔFs after the first addition of Thap. RH versus Δ: $p = 0.0005$ and CM versus Δ: $p < 0.0001$; RH: $n = 7$, $N = 4$; Δ: $n = 7$, $N = 4$; CM: $N = 3$. Data are presented as mean ± SD. **h** Quantification of the ΔFs after the first addition of Ca²⁺. RH versus Δ: $p = 0.445$; CM versus Δ: $p = 0.273$. RH: $N = 3$; Δ: $n = 7$, $N = 3$; CM: $N = 3$. Data are presented as mean ± SD. **i** Quantification of the ΔF after the addition of Ion from experiments similar to the ones in (**e, f**). RH versus Δ: $p = 0.003$; CM versus Δ: $p = 0.0021$. RH: $N = 3$; Δ: $n = 4$, $N = 3$; CM: $n = 4$, $N = 3$. ΔF/F₀ represents the change in fluorescence relative to the basal fluorescence. Data are presented as mean ± SD. $p$ value: unpaired two tailed t test performed in all comparisons.

and bars in g), the addition of Thap resulted in an increase in apicoplast Ca²⁺. In stark contrast, there was little effect of extracellular Ca²⁺ (Fig. 7e, f). These data indicate that the apicoplast selectively sequesters Ca²⁺ released from the ER.

We next tested the response of the *ΔTPC-a* mutant to Thap and Ca²⁺ (Fig. 7b, c, *blue traces and bars*). The cytosolic Ca²⁺ increases after Thap or Ca²⁺ were unchanged in the *ΔTPC-a* mutants indicating that ER Ca²⁺ release and Ca²⁺ influx were not affected by TgTPC deletion (Fig. 7b–d, *blue traces and bars*).

However, the Thap-induced Ca²⁺ signals in the apicoplast were almost absent in the *ΔTPC-a* mutants (Fig. 7e, f, compare *dark gray* and *blue traces* after addition of Thap, quantifications shown in Fig. 7g). The quantification of the *ΔTPC-a* mutant response to extracellular Ca²⁺ is shown in Fig. 7h. The ionophore ionomycin which causes Ca²⁺ release from the ER[37] and other organelles also caused an increase in apicoplast Ca²⁺ uptake in the RH, which was also significantly reduced in the *ΔTPC-a* (Fig. 7e-f, quantification shown in Fig. 7i). The phenotype of the *ΔTPC-a*

mutant was not due to defective ER $Ca^{2+}$ handling because ER $Ca^{2+}$ uptake and leak was indistinguishable between the $Tati\Delta ku80$ and $\Delta TPC$-a mutants (Supplementary Fig. 10d). This experiment was done using digitonin-permeabilized parasites previously loaded with the low-affinity $Ca^{2+}$ indicator MagFluo4, which compartmentalizes into and reports $Ca^{2+}$ in the ER lumen[38].

To assess the specificity of the $Ca^{2+}$ handling phenotype seen in the TPC knockout, we complemented the $\Delta TPC$-FNR-GCaMP6-HA mutant with the $TgTPC$ gene ($\Delta TPC$-TPC-FNR-GCaMP6). The Thap and the ionomycin induced $Ca^{2+}$ signals were recovered to the same levels as the parental in the complemented mutant (Fig. 7e-i, green traces and bars).

These results identify the apicoplast as a $Ca^{2+}$-signaling organelle and reveal a physiological role for TgTPC in regulating selective take up of $Ca^{2+}$ from the ER.

**TgTPC regulates contact site formation between the endoplasmic reticulum and the apicoplast.** Inter-organelle transfer of $Ca^{2+}$ is often ascribed to $Ca^{2+}$ microdomains formed by membrane contact sites[39]. We therefore performed super-resolution microscopy to probe physical interaction between the ER and the apicoplast. Parasites were transfected with p30-GFP-HDEL[40], which labels the endoplasmic reticulum with GFP and IFA performed using antibodies to GFP and the plastid ubiquitin-like protein (PUBL)[41]. This analysis showed the proximity of both organelles especially noticeable at specific extensions of the ER membranes (Supplementary Fig. 11, arrows).

To more directly probe proximity, we performed electron microscopy. As shown in Fig. 8a (upper panels), we resolved numerous apicoplast-ER contact sites in $Tati\Delta Ku80$ cells (small arrows). Recent studies in mammalian cells identified a role for TPCs in regulating contact site-formation between late-endosomes and the ER[42]. We therefore compared apicoplast-ER

contact sites in $Tati\Delta Ku80$ (Fig. 8a upper panels), $\Delta TPC$ (Fig. 8a, middle panels) and complemented $\Delta TPC$-a-TPC parasites (Fig. 8a, bottom panels) cells. As summarized in Fig. 8b, TPC depletion decreased the % of apicoplasts in contact with ER by ~3-fold. To further analyze this defect, we quantified the minimum distance between the organelles. This distance was significantly increased in the TPC null cells (Fig. 8b, right). Both effects were rescued by TPC expression. Immuno-EM localization of TPCs in TPC-3HA cells (Fig. 8c, arrows) revealed the presence of TgTPC at or near the contact site.

In summary, our results show that the apicoplast is physically as well as functionally coupled to ER $Ca^{2+}$ stores through TgTPC.

## Discussion

We report the presence and functional role of the *T. gondii* two-pore channel (TgTPC). The channel localizes to the apicoplast and it is important for tachyzoite growth. Conditional silencing of TgTPC impacted all the steps of the *T. gondii* lytic cycle including host cell invasion, replication, motility and egress. Additionally, apicoplast morphology was altered and its essential functions were repressed. Interestingly, the phenotypic examination of null TPC mutants exposed the main impact of TgTPC in parasite replication while the other defects observed with conditional silencing of the gene resulted from apicoplast dysfunction. The severe growth phenotype caused by the deletion of TgTPC was fully complemented by the expression of TPC and the predicted pore domain was essential for the function of TgTPC. Based on the complementation and its reversibility by conditional silencing of the complementing copy of TgTPC, we propose a role for TgTPC in maintaining the integrity of the apicoplast and its biogenesis. We demonstrate for the first time the presence of $Ca^{2+}$ in the apicoplast and the involvement of TgTPC in its uptake and find a role for TgTPC in maintaining contact with the ER to facilitate this.

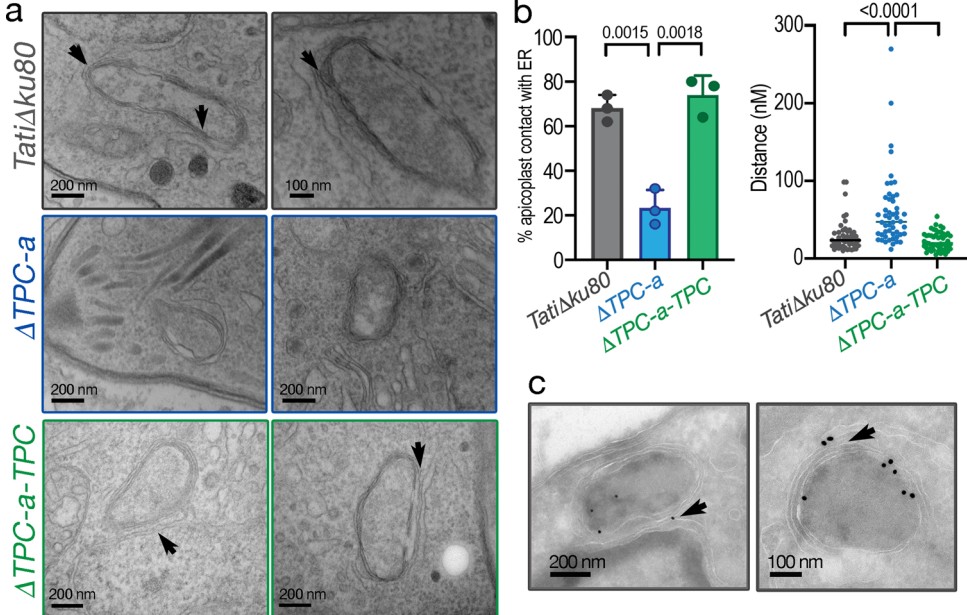

**Fig. 8 TgTPC regulates contact site formation between the endoplasmic reticulum and the apicoplast. a** Representative ER contact sites in Tati$\Delta$ku80 cells (upper panels). The close contacts are indicated with the small arrows. Middle panels show images from the $\Delta TPC$-a mutants. Lower panels show images of the $\Delta TPC$-a-TPC complemented mutant. **b** Statistical analyses of the contact sites. The percentage of apicoplasts that are in close contact with the ER ( < 30 nm) in three technical repeats in different days were analyzed for comparison between the three cell lines (left panel). $\Delta TPC$-a versus Tati$\Delta$ku80: $p = 0.0015$; $\Delta TPC$-a versus $\Delta TPC$-a-TPC $p = 0.0018$. The ER-apicoplast distance in 50 randomly chosen cells was analyzed (right panel). $p < 0.0001$. Data are presented as mean ± SD. Student's *t* test. *p* value: unpaired two tailed *t* test performed in all comparisons. **c** Immuno-EM of *TPC-3HA* in i$\Delta TPC$-3HA cells with αHA antibody. The TPC signal was often found in ER-Apicoplast contact sites (arrows). N = 2.

As a product of a secondary endosymbiosis process, the apicoplast is surrounded by membranes derived from the cyanobacterium that gave origin to the plastid, the endosymbiotic alga, and the host endosomal compartment[21]. Metal ions play essential roles for the activity of more than one-third of all resident enzymes and considering the large number of essential activities present in the apicoplast, it is likely that ions will be highly regulated in the organelle. No regulators of $Ca^{2+}$ levels in the apicoplast have been reported until now. The discovery of a TPC localized to the apicoplast is therefore a unique finding. Although our data identify pore-dependent defects in growth and disrupted $Ca^{2+}$ homeostasis, direct electrophysiogical characterization of TgTPCs is warranted. One essential apicoplast metabolic pathway of cyanobacterial origin is the 1-deoxy-D-xylulose-5-phosphate (DOXP) pathway, which synthesizes the five-carbon precursor of isoprenoids, isopenthenyl diphosphate (IPP)[34]. IPP was found to be the most critical product of the apicoplast in some life stages of the malaria parasite[22,43]. In this regard, interfering with other aspects of apicoplast maintenance or metabolism would also impact the ability of the organelle to supply IPP. In our experiments, Geranylgeraniol (GGOH), the downstream product of IPP partially rescued the retarded growth of the TgTPC *null* mutants. It could not fully replace IPP, as it could be needed for other functions upstream the production of GGOH like farnesylation of proteins. This could explain the partial rescue by GGOH.

TPCs function as $Ca^{2+}$ release channels in animal cells and are activated by the second messenger nicotinic acid adenine dinucleotide phosphate (NAADP)[6–8]. This could be the case for TgTPC. It has been demonstrated that ADP-ribosyl cyclases, which generate NAADP and cyclic ADP ribose[44] are present in tachyzoites[45]. Animal TPCs also function as $Na^+$ channels when activated directly by PI(3,5)$P_2$ (phosphatidylinositol 3,5-bisphosphate)[12,46,47]. In this regard, it was proposed that PI(3,5)$P_2$ might be synthesized at the apicoplast membrane by PIKfyve[48,49] to maintain the morphology of the apicoplast[50].

Our direct $Ca^{2+}$ measurements within the apicoplast lumen identify the apicoplast as a novel $Ca^{2+}$ store. We show that $Ca^{2+}$ release from the ER but not $Ca^{2+}$ influx results in $Ca^{2+}$ uptake into the apicoplast. This highly selective transfer suggests a tight coupling between $Ca^{2+}$ release and $Ca^{2+}$ uptake and is reminiscent of ER-mitochondrial coupling[51]. Our identification of contact sites between the ER and the apicoplast would facilitate communication between the two. Similar contacts have been identified in *T. gondii*[52] and *Sarcocystis*[53]. Importantly, we show that deletion of TgTPC disrupts $Ca^{2+}$ uptake and contact. TgTPC therefore appears to regulate membrane contact site formation similar to the role reported for TPC1 at ER-endosome junctions in mammalian cells[42]. In this scenario, TgTPC would facilitate $Ca^{2+}$ uptake through a second transporter at the contact site. Alternatively, TgTPC might directly mediate $Ca^{2+}$ uptake akin to the mitochondrial $Ca^{2+}$ uniporter at ER-mitochondria junctions. This model would necessitate a lumen negative membrane potential. How uptake of $Ca^{2+}$ into the apicoplast multi-membrane system is transmitted to the lumen requires investigation. Regardless of the exact mechanism, communication between the apicoplast and the ER is essential for apicoplast function and cell growth and identifies TgTPC as a novel protein worth exploring as a potential target for combatting Toxoplasmosis.

## Methods

**Phylogenetics**. Multiple sequence alignment and phylogenetic reconstruction. Protein sequences were aligned using either MAFFT or T-Coffee and columns containing more than 50% gaps were subsequently removed from the sequence alignment using GapStreeze (Gap Strip/Squeeze v 2.1.0). Unambiguous sequence alignments were then converted to PHYLIP or NEXUS format. Identifiers for metazoan and unicellular TPCs are listed in Supplementary Table 1. ProtTest was utilized to select the best-fit evolution model and parameter estimates for the phylogenetic analyses. Maximum likelihood phylogeny with 100 bootstrap replicates was performed using PHYML (version 3.1) with the LG amino acid substitution model, estimated proportion of invariable sites, empirical amino acid frequency estimation, and the four-category discrete gamma model (LG + I + G + F) selected by ProtTest (Version 3.4.2)[54,55]. Consensus trees were obtained using the CONSENSE program from the PHYLIP package (Version 3.69).

**Structural modeling**. Domain architecture of TPCs was predicted using Batch CD-Search[56]. Sequences corresponding to the pore region of domain I and domain II of TgTPC1 were submitted to iTASSER and the top models for each assembled as a tetramer based on the structure of Arabidopsis TPC (pdb:5dqq). The tetramer was locally minimized using the YASARA NOVA forcefield[57] and presented using PyMOL.

**Toxoplasma growth**. *T. gondii* tachyzoites (RH strain, *Tati∆ku80*) were cultured in hTERT human fibroblasts[58] using Dulbecco's modified minimal essential media (DMEM) with 1% fetal bovine serum. Extracellular tachyzoites were collected from cultures containing 50–75% of egressed parasites. Intracellular tachyzoites were manually released by scrapping off and passing the suspension through a 27 G needle. Parasites (both intracellular and extracellular) were purified as described[19]. Electroporation was performed using a Gene Pulser Xcell from BioRad. Stable transfectants were selected with 20 µM chloramphenicol or 1 µM pyrimethamine and cloned by limiting dilution. The *Tati∆ku80* mutant, was a gift from Dr. Boris Striepen.

**Generation of mutants**. The *TgTPC* cDNA was amplified with primers **1** and **2** (Supplementary Table 4) (underlined nucleotides correspond to restriction sites for BglII and AvrII). The PCR product was cloned in the Zero Blunt TOPO cloning vector, followed by sequencing and subsequent cloning into the pDTM3 vector[19]. For in situ tagging, a fragment of approximately 2 kbs was amplified from the genomic locus (3′ region) of the *TgTPC* gene using primers **3** and **4** (Supplementary Table 4). The fragment was cloned in the pLic-3HA-CAT plasmid[24] and the construct was linearized with SphI for transfection of *Tati∆ku80* parasites. Clonal cell lines were generated after selection and subcloning and termed *TPC-3HA*. A promoter insertion plasmid was generated by cloning two fragments from the 5′ end of the *TgTPC* gene into the pDT7S4myc plasmid[24]. One fragment corresponds to the *TgTPC* 5′ flanking region (predicted promoter/5′UTR) and was amplified with primers **5** and **6** (underlined sequences correspond to NdeI restriction sites). The second fragment corresponds to the 5′ *TgTPC* coding sequence beginning with the start codon, which was amplified with primers **1** and **7** (Supplementary Table 4) (underlined sequences correspond to BglII and AvrII restriction sites). The plasmid was linearized with AvrII for transfection of *Tati∆ku80* and *TPC-3HA* cells. The clonal lines created after selection and subcloning were termed *i∆TPC* and *i∆TPC-3HA*.

Cosmid **PSBLZ79**, which contains the *TgTPC* genomic locus was obtained from Dr. L. David Sibley (Washington University). The knockout cassette (chloramphenicol acetyl-transferase (CAT) gene for *T. gondii* and gentamicin for *E. coli* selection) from pH3CG[59] was amplified with primers **8** and **9** (Supplementary Table 4). The *TgTPC* gene targeting cosmid construct was made by replacing ~ 5 kb of the 5′ end of the *TgTPC* gene with the knockout cassette. This was done by recombineering in *E. coli* EL250 as described previously[31]. The *TgTPC* targeting cosmid was then electroporated into *i∆TPC-TR* cells. After chloramphenicol selection several *∆TPC* clones were isolated and one was selected for further studies: *∆TPC-a*. For complementation of *∆TPC-a* cells, a plasmid pDTM3-TPC was used. The introduced cDNA was confirmed by PCR with primers **1** and **10** that amplify a 2.4 kb cDNA fragment. These two primers failed to amplify *T. gondii* genomic DNA because of the presence of introns. Primers **11** and **12** were used as controls to amplify the 5′UTR region from wild type parasites. Supplementary Table 2 lists all the cell lines created in this study and a detailed explanation of how the ∆TPC mutant was generated.

**Complementation with a conditional copy of *TgTPC***. The cDNA of the *TgTPC* was cloned in the Zero blunt Topo vector from Invitrogen. Mutations in the pore domains of the protein were generated with a GeneArt® Site-Directed Mutagenesis System from Invitrogen. The oligonucleotides used for mutations in the Pore 1 domain were **13** and **14**. The oligonucleotides used for mutations in the Pore 2 domain were **15** and **16**. Mutations were verified by sequencing. Both wild type and pore mutants were cloned in the vector pDT7S4myc[24] for overexpression in *T. gondii*. The pDT7S4myc vector contains a regulatable element that allows the regulation of the downstream gene with addition of ATc. This strategy works only if the construct is introduced into parasites that express the tetracycline transactivator.

**Southern blot analyses and apicoplast DNA quantification**. *T. gondii* genomic DNA was purified and digested with EcoRI for Southern blot analysis. The probe against the coding sequence of the *TgTPC* gene was generated by PCR with primers 1 and 17. The probe against the 3′-UTR was amplified with primers 18 and 19. The

probe used for apicoplast DNA was generated by using primers: 20 and 21[60]. For nuclear DNA the probe used was generated against the *TgFPPS* gene[59]. For quantification of the apicoplast DNA, purified PCR products were labeled with $^{32}$P by random priming. Southern blot signals for apicoplast DNA and nuclear DNA (TgFPPS) in the same membrane were quantified by PhosphorImager and ImageQuant software.

**Whole-genome sequencing of ΔTPC mutants**. Genomic DNAs from the mutant strains was extracted by using the quick DNA miniprep plus kit from Zymo Research, and sent to BGI (Beijing Genomics Institute) genomics (San Jose, CA, USA) for whole-genome sequencing. The genomic DNA of the *iΔTPC-TR, ΔTPC-a* and *ΔTPC-b* cells was extracted from frozen stocks prepared soon after the mutants were generated. These cells have been cultured in vitro for a much shorter time than the mutant used for the analysis presented in Supplementary Fig. 5d. The whole-genome re-sequencing (WGRS) of the *T. gondii* mutants (3 Gb of Sequencing per library for about 37X coverage) and the bioinformatics analysis of the sequencing data were performed in BGI headquarters (ShenZhen, China).

**Growth analysis**. Plaque and growth assays were performed as described[59]. All mutant lines (*iΔTPC-3HA, iΔTPC, ΔTPC* and complemented cells) were transfected with a plasmid containing a tandem dimer Tomato (tdTomato) gene[34]. Red parasites were enriched by FACS sorting and subcloned by limiting dilution. A standard curve was developed to calibrate fluorescence levels to number of parasites for each parasite line. For replication experiments, hTERT cells were grown on 35 mm Mattek dishes and each dish infected with 50,000 tdTomato-expressing parasites. 24 h after infection counting parasites per PV was done in a fluorescence microscope. For each experiment, at least 100 PVs were counted. Results were the average of 3 independents experiments. For egress assays, monolayers of hTERT cells grown in 35 mm Mattek dishes were infected with 50,000 tdTomato-expressing parasites. Egress was triggered with 1 μM ionomycin using 24 h old cultures. The invasion was evaluated using a red-green assay[61]. $2.5 \times 10^7$ tachyzoites were added to subconfluent hTERT monolayers and allowed to settle for 15 min on ice, then incubated for 2 min at 37 °C. Subsequently, the medium was aspirated and fixed with 2.5% formaldehyde for 20 min. Subsequent steps were done as published[61]. Three independent experiments were done, each one by triplicate, counting 8 randomly selected fields per individual coverslip. The number of invaded parasites was calculated as the difference between the total number of parasites (*green*) and the number of attached parasites (*red*). Results are expressed as means ± S.D. Statistical analysis was performed using the Student's *t*-test. Differences were considered significant if *P*-values were < 0.05.

For migration experiments, freshly lysed parasites were filtered and incubated in ringer buffer at 37 °C. $5 \times 10^6$ parasites were added to BD Biosciences membranes containing 200 μL of the same solution in the upper chamber and 500 μL in the lower chamber. Parasites were allowed to migrate through the 3 μm pore and samples from the lower chamber were taken at 5, 10, 15, 30, 60, 90, and 120 min for counting. Negative controls were done pre-incubating with 50 nM bafilomycin A or 500 nM cytochalasin D for 15 min.

**Microscopy and western blot analyses**. Tachyzoites were grown on hTERT cells on cover slips for ~ 24 h, washed twice with buffer A with glucose (BAG, 116 mM NaCl, 5.4 mM KCl, 0.8 mM MgSO$_4$, 50 mM HEPES, pH 7.2, and 5.5 mM glucose) and fixed with 4% paraformaldehyde for 1 h, followed by permeabilization with 0.3% Triton X-100 for 20 min, and blocking with 3% bovine serum albumin. IFAs were performed as previously described[19]. Fluorescence images were collected with an Olympus IX-71 inverted fluorescence microscope with a Photometrix CoolSnapHQ CCD camera driven by DeltaVision software (Applied Precision, Seattle, WA). Super-resolution microscopy was performed using a Zeiss ELYRA S1 (SR-SIM) microscope with a high-resolution Axio observer Z1 inverted microscope stand with transmitted (HAL), UV (HBO,) and high-power solid-state laser illumination sources (405/488/561 nm), a 100x oil immersion objective, and an Andor iXon EM-CCD camera and acquired using the ZEN software (Zeiss) with a SIM analysis module. Rat anti-HA antibody from Roche was used at a dilution of 1:25. Mouse anti-HA antibody from Covance was used at a dilution of 1:200. The Rabbit anti-hsp60 antibody was used at 1:1,000 and the rabbit anti ACP antibody was used at a dilution 1:200. Both apicoplast antibodies were a gift from Boris Striepen. Mouse anti c-myc antibody was from Sigma and was used at a dilution of 1:500.

For immunoelectron microscopy, freshly isolated parasites were fixed with 4% paraformaldehyde, 0.05% glutaraldehyde in PBS, pH 7.4, on ice for 1 h. After fixation, the parasites were pelleted and gently resuspended in PBS. Parasite suspensions were sent to Washington University MicTocopy service for analysis. Dr. Wandy Beatty performed the cryo immune EM analysis as described[19]. Western blot analysis was performed as previously described[62]. Rat anti-HA antibody from Roche was used at a dilution of 1:200. Mouse anti-HA antibody from Covance was used at a dilution of 1:1,000. Secondary goat anti-rat or mouse antibody conjugated with HRP was used at 1:5,000. Mouse anti-α-tubulin at a dilution of 1:5,000 was used for loading control. Anti-lipoylated-PDH-E2 antibody (clone3H-2H4), was a gift from Dr. Eric Gershwin, UC Davis[63] and was used at 1:1,500 dilution. Apicoplast function was evaluated as described previously[31].

**Calcium measurements**. Parasites were purified and loaded with Fura 2-AM as described[64,65]. An Hitachi F-7000 spectrofluorometer was used for calcium measurements. Conditions and calibration were as published[65,66].

For ER calcium measurements, cells were resuspend to a final density of $1 \times 10^9$ cells/ml in HBS buffer (135 mM NaCl, 5.9 mM KCl, 1.2 mM MgCl$_2$, 11.6 mM HEPES, 1.5 mM CaCl$_2$, 11.5 mM glucose) containing 1 mg/ml BSA, 0.2 mg/ml of pluronic F127 and 20 μM Mag-Fluo-4-AM. Loading was for 60 min in the dark at room temperature. Subsequently, parasites were washed 3 times and the cell pellet resuspended in 1.8 ml CLM (20 mM NaCl, 140 mM KCl, 20 mM PIPES, pH 7.0) containing 1 mM EGTA at $1 \times 10^9$ cells/ml. Cells were permeabilized with 30 μg/ml digitonin for 5 min. Permeabilized cells were washed 3 times with CLM containing 1 mM EGTA and resuspended to a final density of $1 \times 10^9$ cells/ml and kept in ice. For each test, 50 μl ($5 \times 10^7$) of the suspension was added into 1.95 ml of CLM containing 1 mM EGTA with 375 μM CaCl$_2$ (220 nM free Ca$^{2+}$).

**Targeting genetically encoded calcium indicator GCaMP6f into the apicoplast**. The N-terminal targeting sequence of the apicoplast protein ferredoxin reductase (FNR) (150 aa) fused with the GCaMP6f gene was used for apicoplast targeting[67]. The FNR sequence was amplified by primers 22 and 23 (Supplementary Table 4). The GCaMP6*f* gene was amplified by primers 24 and 25. The PCR products were purified and the construct created by overlapping PCR which was cloned into pCR-Blunt II-TOPO vector. The sequence was verified by sequencing and the FNR-GCaMP6 fragment was digested with BglII and AvrII and cloned into the same restriction sites of pDT7S4H3[68]. This final plasmid was electroporated into RH and ΔTPC-a parasites. After selection with pyrimethamine, parasites were FACS sorted and subcloned. Expression of FNR-GCaMP6f in both cells was verified by IFA and western blot analysis (Fig. 7a and Supplementary Fig. 10a). To complement the TgTPC gene in the parasites expressing FNR-GCaMP6f, we transfected the plasmid pDTM3-TgTPC into the ΔTPC-FNR-GCaMP6 mutant. After culturing for 18 days, the transfected cells were subcloned. TgTPC complemented clones were verified by PCR. This clone was named ΔTPC-TPC-FNR-GCaMP6 (or CM in Fig. 7). The parasites expressing *FNR-GCaMP6* were collected, washed, and suspended in Buffer A with glucose at a density of $10^9$/ml. 50 μl of parasites were transferred into a cuvette with 1.95 ml of Ringer buffer (no calcium) containing 100 μM EGTA$_2$. Fluorescence of FNR-GCaMP6 parasites was recorded in a Hitachi F7000 fluorescence spectrometer.

**Transmission electron microscopy**. For ultrastructural observations of intracellular *T. gondii* by thin-section transmission electron microscopy (EM), infected human foreskin fibroblast cells were fixed in 2.5% glutaraldehyde in 0.1 mM sodium cacodylate (EMS) and processed as described[69]. Ultrathin sections of infected host cells were stained before examination with a Hitachi 7600 EM under 80 kV. For quantitative measurement of distance between organelles, the closest point between *T. gondii*'s apicoplast and ER membrane was measured using a line measurement tool. A distance between membranes of 30 nm or less was defined as membrane contact. The first 50 parasites with a visible ER and apicoplast were used for quantification. Statistics calculated from three technical replicates. Quantification was performed following a blinded setup.

**Statistics**. Statistical analyses were performed by Student's *t*-test using GraphPad PRISM version 8.2. Error bars shown represent mean ± SD (standard deviation) of at least three independent biological replicates.

**Reporting summary**. Further information on research design is available in the Nature Research Reporting Summary linked to this article.

## Data availability
The sequencing data generated in this study have been deposited in the NCBI database under accession code PRJNA695295. The data is available for public access. All other data generated or analyzed in this study are included in the article (and its supplementary information files). Source data are provided with this paper.

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

## Acknowledgements

The authors thank Wandy Beatty for the immunoelectron microscopy, Rodrigo de Paula Baptista for helping with the analysis and submission of the sequencing data. The super-resolution microscope is part of the biomedical microscope core (BMC) located at the Coverdell Center for Biomedical Research, UGA. We thank Julie Nelson from the CTEGD Cytometry Shared Resource Lab for expert help with cell sorting. MoFlo XDP (Beckman Coulter, Hialeah, Florida) and S3 (Bio-Rad, Inc., Hercules, California) were used to sort the cells in this study. Boris Striepen, Vern Carruthers, Eric Gershwin, and John Boothroyd provided antibodies. This work was supported by the U.S. National Institutes of Health grants AI096836 and AI128356 to S.N.J.M. X.C. was supported in part, by the UCLA Specialty Training and Advanced Research (STAR) fellowship program and the NIH T32 training grant T32HL007895. T.R. was funded by a fellowship from the Royal Society, UK. S.V. was partially supported by a fellowship through the NIH T32 training grant T32AI060546. I.C. was supported by the NIH grant R01AI060767. S.P. was funded by BBSRC grants BB/N01524X/1and BB/T015853/1.

## Author contributions

Z.L. performed and coordinated most of the experiments, analyzed the data, wrote the manuscript; T.P.K. performed some initial experiments and analyzed data; L.A. performed initial experiments; B.A. electron microscopy experiments and analysis; X.C. and T.R. phylogenetic, modeling and pore analysis; S.A.V. performed SIM experiments; I.C., EM analysis, S.P., writing, review, editing, analysis and interpretation of data; S.N.M. coordinated the project and experiments, contributed resources and wrote the manuscript.

## Competing interests

The authors declare no competing interests.
