## [Peer Review File · Nature Communications]

Reviewer #1 (Remarks to the Author):

In their manuscript, Li et al describe a novel two-pore channel (TPC) that localises to the apicoplast of human parasite *Toxoplasma gondii*. Due to its endosymbiotic origin and the fact that it harbours essential metabolic pathways, investigating the apicoplast is often cited as a privileged avenue to search for potential drug targets. It is bound by four membranes and likely undergoes extensive exchanges with other subcellular compartments, yet very few transporters or channels have been identified so far. In this regard, this manuscript is thus potentially interesting, although as mentioned by the authors this channel is only present in Coccidian parasites.

The authors have generated a large amount of data to unambiguously demonstrate that TgTPC is involved in maintaining the integrity of the apicoplast and they also bring evidence that the organelle contains Ca²⁺. In the introduction and the discussion, the authors put forward TgTPC as a potential drug target against toxoplasmosis. A number of essential apicoplast proteins have been identified in the past and have yet to be validated as bona fide druggable targets. Not only the manuscript currently provides no clear demonstration as to why TgTPC is essential for apicoplast homeostasis or cell replication (regulating Ca²⁺ availability in the organelle for signalling purposes or providing a co-factor for essential processes?), but the prospect of identifying or designing specific inhibitors of this Ca²⁺ channel may also appear as an unrealistic task as it stands. Another main concern I have about this manuscript is the extensive use for phenotypic analysis of a knock-out cell line that has been kept of about a year in culture and has shown striking changes in organelle recovery and growth. I understand the authors took care in providing complemented strains to verify the specificity of the phenotypes, but given the high adaptability of *T. gondii* (and of microorganisms in general) in cultures, it is possible that this cell line is different in many regards from the initial mutant and hence may not be completely adequate to draw clear-cut conclusions.

Specific points.

As a general comment, please provide line numbers when formatting the manuscript, it makes it much easier for the reviewers to refer to specific points.

Some parts of Fig. 1 (ie conservation of the pore sequence, domain arrangement) are somewhat redundant with the information available in the Prone and Taylor 2011 PloS One paper.

It may have been useful to assess for overall solubility of TgTPC, as it would be quite complementary to the localisation at the membrane(s) of the apicoplast.

“Somewhat surprisingly, we found little overlap of staining with TgCPL, a marker of the Plant-Like Vacuole (PLV)/VAC” : when discussing TgTPC localization in the corresponding results section, it may be useful to remind the reader of the vacuolar localization of its homolog in higher plants. Also, there is no such thing as ‘data not shown’ these days, please show PLV colocalization as a Suppl. Figure.

pp.4-5. After long-term incubation with ATc, the authors state that “All steps of the lytic cycle therefore were affected TPC in the Apicomplexan parasite *T. gondii* following prolonged-ATc treatment, likely the result of apicoplast fragmentation and its consequent dysfunction.”. Maybe they could be more explicit, as it quite simply seems to reflect a loss of viability.

Related to that point, on p. 8 they write “Alterations in the apicoplast, as those we report here for the TgTPC conditional mutants, are numerous and a clear demonstration of the essentiality of the apicoplast metabolic pathways for *T. gondii* motility, host invasion and egress”. This is in contradiction with the existing literature. It is commonly accepted that apicoplast loss leads to the so-called ‘delayed death’, by which parasites that have just experienced the loss of the organelle are nonetheless able to egress and invade new hosts, and only upon invading a new host cell they will be unable to replicate without the organelle. In fact, it is strange the term ‘delayed death’ is

not even mentioned in this manuscript.

p.5. "they stop responding to ATc". This is confusing as down-regulation still seems effective.

As mentioned in my general comments, I am wondering if there might be another plastidic ion transporter that would be mutated/upregulated in the adapted TgTPC cell line to perform a similar function as TgTPC, albeit sub-optimally (just enough to recover some apicoplast functions, but not enough to recover the wild-type phenotypes, which would be only fully restored after complementation with TgTPC). Whole genome sequencing and/or comparative transcriptomic analysis of the initial and the adapted mutant cell line may provide insights into this.

I am convinced by the Ca²⁺ probe use and measurements. I would just advise the authors to avoid drawing definitive conclusions such as "Thap-induced Ca²⁺ signals in the apicoplast were absent in the DTTPC-a mutants", as there seems for instance to be a small response on the right graph (when thapsigargin was added after Ca²⁺ load). Also for Fig.7, as several conditions (order for adding Thap and Ca²⁺) were used, it is not clear to which ones the delta Ca²⁺ and deltaF bar graphs correspond.

Out of curiosity, is it possible to estimate the amounts of Ca²⁺ in the apicoplast? I suppose it may be relatively low as this group previously used elemental mapping and electron microscopy to detect calcium in *T. gondii* tachyzoites and acidocalcisomes seem to represent the main Ca²⁺ store. Related to this, the GCaMP6 showing discernible fluorescence change by microscopic observation? If so, can it be used for live imaging to check if there are Ca²⁺ variations in the apicoplast at specific moment of the cell cycle?

Maybe a bit outside the scope of the study, but as there is no TPC homolog in Plasmodium, it would be interesting to express an apicoplast Ca²⁺ sensor in this Apicomplexa (and it would be quite compelling to find out there is none).

As the super resolution microscopy data do not bring additional dynamic and kinetic insight into the ER/apicoplast contact sites, it is not more informative than what was already described (and rightfully referred to by the authors in the discussion) by the electron tomography analysis of Tomova et al Traffic 2009. Thus it may be moved to supporting information.

NAADP is a known activator of TPC channels for Ca²⁺ flux in animal cells, while this is extensively discussed by the authors, it is frustrating this could not be tested to see if can modulate Ca²⁺ in the *T. gondii* GCaMP6 reporter line. Although NAADP is largely membrane impermeable, several publications show the use of caged compounds (Churchill et al 2002) or even a cell permeant version of NAADP (Parkesh et al Cell calcium 2008), thus I wonder if these may be used in *T. gondii*.

Reviewer #2 (Remarks to the Author):

This interesting study examines the role of a putative two-pore ion channel in the biology of *Toxoplasma gondii* parasites. As a whole, the paper presents a solid analysis of the role of this candidate ion channel (termed TgTPC) in the apicoplast, a plastid organelle found in *T. gondii* and related parasites. The authors demonstrate the localization of this putative channel to the apicoplast periphery and undertake a comprehensive analysis to demonstrate that the protein is important for parasite growth, where it plays a critical role in apicoplast maintenance. The paper uses an interesting approach to knockout TgTPC in a parasite strain pre-adapted to its depletion, and presents some evidence that loss of TgTPC results in defects in Ca²⁺ translocation into the apicoplast. Aspects of the paper require further experimentation to support the overall conclusions, in particular to address the adaptations that have occurred in response to TgTPC knockdown.

Key major comments:

1. Figure 5 and associated experiments. The approach to generate a TgTPC knockout in a strain adapted for growth following TgTPC depletion is a novel one. This enables the authors to measure

the effects of TgTPC knockout without a concurrent loss of the apicoplast. Presumably, there has been some genetic/epigenetic/metabolic changes that have occurred in these parasites to enable the adaptation to growth in the absence of TgTPC. The authors present no data on what these changes are. Without a clearer explanation of the adaptation that has occurred in these parasites, the subsequent (indirect) experiments analysing the effects of TgTPC on Ca²⁺ changes in the apicoplast are difficult to interpret. Are these changes due to loss of TgTPC, or to adaptations that have occurred in this strain, or (perhaps most likely) a combination of the two? The paper would be strengthened considerably if the authors provided an explanation for what has changed in parasites following long-term culture of the TgTPC knockdown strain. They could perhaps undertake whole genome sequencing to detect adaptive mutations in the genome or RNAseq analysis to measure transcript abundances to better characterize the adaptations that have occurred. I suspect these experiments will take some time to complete and follow up on.

2. Figure 7. The approach to using Ca²⁺-sensitive GFP fluorescence to examine Ca²⁺ concentrations in the apicoplast is, in principle, a powerful one. As noted in the previous comment, the approach used in generating the TgTPC knockout mutants make it difficult to distinguish between the effects of TgTPC loss and parasite adaptations to TgTPC loss. The authors could consider undertaking control GCaMP6 experiments in both the original Tet-regulatable strain complemented with WT TgTPC and the knockout strain complemented with WT TgTPC to determine whether there are any differences in apicoplast Ca²⁺ responses in parasites expressing TgTPC that have either not adapted (original Tet strain) or adapted (complemented knockout strain) to loss of TgTPC. This could reveal whether adaptation changes overall apicoplast Ca²⁺ responses or not.

Other comments that require further experimentation:

3. Figure 3C. The apicoplast “appeared swollen”, “fragmentation was confirmed by electron microscopy”. The authors present a single image that claims to show an apparently abnormal apicoplast. This image does not convincingly show either a swollen or a fragmented apicoplast. Without a thorough quantitative analysis of apicoplast morphology upon TgTPC depletion, including measurements of the dimensions of a normal apicoplast and how this changes upon TgTPC depletion, I don’t think the authors can make this claim.

4. Figure 4E. Is the defect in ionomycin-dependent egress due to the specific role of TgTPC in apicoplast/parasite biology, or because of apicoplast loss that occurs as a consequence of TgTPC knockdown, and which earlier experiments show is probably substantial after 5 days on ATc? The authors should undertake a more thorough experiment, including (at a minimal) some earlier time points where TgTPC knockdown is substantial (e.g. day 2 post-ATc addition, Fig. 2C), but where some of the morphological defects may be less pronounced (see minor comment 9 below).

5. Page 5. “ΔTPC-a mutants were further cultured for several months, which resulted in gradual recovery of apicoplast labeling with markers like ACP (Fig. 5B).” Without some sort of quantification here, I don’t understand the basis for this statement. Apicoplasts can only arise from pre-existing apicoplasts. Is it that the loss-of-apicoplast phenotype disappears over time?

6. Figure 6. The authors show that the constitutive expression in the ΔTgTPC strain of mutant versions of TgTPC lacking key Lys residues in the TPC pore fail to rescue growth, and interpret this as evidence that channel activity in TgTPC is required for this protein to carry out its function. An alternative explanation is that the mutant TgTPC proteins are not expressed at sufficient abundance to complement growth (e.g. if the mutation causes destabilization and subsequent degradation of the protein). The authors should perform western blotting to determine how the abundance of the mutant TgTPC isoforms compares to abundance of the complementing WT TgTPC protein.

7. Figure 7D-E. The apparent interaction between the apicoplast and ER is intriguing, and something that has been noted in previous studies. However, the authors conclusions are based on only three images. If they want to include these data, they should quantify the extent to which the ER and apicoplast interact.

Minor comments:

Introduction

1. Page 2. "But TPCs are also activated by the endosomal phosphoinositide". The phrasing is confusing here. Are all TPCs activated by PI(3,5)P2? Or only a subset (like TPC1 and 2 from mammals)?
2. Page 2. Is it correct to refer to the apicoplast as a "chloroplast remnant"? It is a plastid that retains numerous canonical plastid functions, so is not really a "remnant" organelle.
3. Page 2. What is the evidence that the vacuole of these organisms is plant-derived in an evolutionary sense? Many other organisms have vacuoles or lysosomes, which appear to be functionally similar to the describe PLV/VAC compartment.

Results and figures

4. Figure 1A. Given the discussion of plant TPCs in the introduction and the link of TgTPC with the apicoplast, the phylogenetic tree would benefit from inclusion of some plant TPCs. Are the sequences of plant TPCs conserved to a sufficient extent to enable them to be included in the phylogenetic analysis?
5. Figure 1A. The tree is missing representatives of the closest free-living relatives to apicomplexans (e.g. chromerids). Do the TPC homologues from chromerids group with the coccidian TPC clade (which would suggest a homologue was present in the common free-living ancestor of apicomplexans and lost in hemosporidians) or do coccidian TPCs form a distinct clade as argued in the text? I think an expanded analysis would add more value to the evolutionary angle of the study.
6. Figure S1. I found the Southern blots difficult to interpret. What are the expected sizes of the bands on this Southern blot upon locus modification to introduce the 3HA and the tetracycline-regulatable element? And do these match to the observed bands? The 3.9 kb band expected for probe 2 in WT parasites in the diagram is the same size as the band observed in the modified locus with probe 1. This raises concerns about the strategy to use probe 2 on a membrane where the signal from probe 1 is still visible. The authors should provide more details on how they verified the genetic modifications.
7. Page 3. "we found little overlap of staining with TgCPL ... (not shown)". Given that TPCs localize to vacuoles/lysosomes in other organisms, perhaps worth showing these data in a supplementary figure?
8. Figure 2E. This image is very zoomed-in. This makes the context of the labelled structures within the parasite cell difficult to establish. The authors should include an image of the entire cell here.
9. Figure S2. The data demonstrate that defects in apicoplast biogenesis are apparent 3 days after ATc addition. Later experiments in the study examine processes such as egress at time points that occur well after these biogenesis defects are apparent. The paper would be strengthened by establishing whether apicoplast loss is concomitant with TgTPC depletion, or whether apicoplast loss occurs subsequent to TgTPC knockdown. This could be achieved by including earlier time points (e.g. 1 and 2 days) in this experiment.
10. Figure 3A and page 4 "the apicoplast marker ACP ...was no longer detectable after culturing Δ TPC-3HA with ATc for 7 days." According to the figure, the day 7 sample was labelled with anti-Hsp60 as the apicoplast marker not anti-ACP.
11. Figure S2 legend. Need to describe what the errors bars represent in these column graphs and what statistical analysis was performed.

12. Figure 3D. The antibody used is not clear. The text in the main body states that it is an anti-lipoic acid antibody, whereas the figure legend states anti-lipoylated PDH-E2. The authors should clarify this. The methods suggests an anti-lipoylated PDH-E2 antibody was used, in which case the main body of the text should be changed.

13. Figure 3D, column graph. The authors should indicate what the error bars depict, and what statistical analysis was performed.

14. Figure 4A-B. Are the growth defects due solely to depletion of TgTPC? These data would be strengthened by inclusion of a strain wherein a constitutively-expressed TgTPC is added back to the regulatable strain to see if this complements parasite growth upon depletion of the regulated TgTPC gene (as the authors do in later experiments in the tetracycline-resistant strain).

15. Figure 4B-F. Need to indicate what the error bars represent.

16. Figure S3B and text. The Tet-resistant strains "expressed neither TgTPC (Fig. S3B) nor ACP (Fig. S3B)". Western blotting is more appropriate than IFAs of measuring whether these proteins are expressed or not, and it would seem unlikely that ACP is not expressed at all in these parasites. Perhaps the more general point to make is that apicoplast appears to be absent in the resistant strain.

17. Figure 5D, F-H. Need to describe what error bars in graphs represent. Where relevant, need to describe statistical analysis that was performed.

18. Page 6 text. "(Fig. 6B middle panels and B)". Remove the "and B"? Do the authors mean to refer to the right image of the middle panel? This needs clarification.

19. Figure S7C. Figure legend needs to describe what the error bars represent. I've noted this elsewhere as well, but in all graphs depicted in the manuscript, the authors must describe what error bars represent (almost always missing), and present a description of the statistics that they carried out (occasionally missing).

20. Figure 6E-F. Given that 6E represents the averaged data from 6F, it makes more sense to show these the other way round, or combined into a single figure.

21. Page 6. "these data indicate that the TgTPC directly impacts *T. gondii* replication and the pore domain of TgTPC is essential for its function". Since they don't have an assay that directly measures TgTPC pore activity (e.g. electrophysiology studies on purified apicoplasts or on TgTPC expressed in a heterologous system), this conclusion is over-reaching what the data show. Rather, the authors should inject a note of caution "... data are consistent with the pore domain of TgTPC being important for its function."

22. Figure 7. Why was RH used as a WT strain here and not TATi Δ ku80 (i.e. the parental strain)?

23. Figure 7C. The authors need to specify what the $\Delta F/F_0$ value on the y axes represent. The changes in apicoplast [Ca²⁺] upon thapsigargin addition in WT parasites are small. Although the authors find statistical significance in the data (ΔF after Thap in WT vs mutant parasites), I think it would help readers if the variation in the data were better represented. Ideally, the authors would plot $\Delta F/F_0$ data points from each individual experiment on the graph rather than representing the data as column graphs. At the very least the authors need to specify what the error bars represent (SD, CI or SEM?).

24. Figure 7C legend. As a second comment on Figure 7C, do the values on the column graphs to the right of the traces represent the ΔF after thapsigargin in the "Thap before Ca²⁺" traces, or a combination of the "Thap before Ca²⁺" and "Ca²⁺ before Thap" traces? The increase in fluorescence following thapsigargin addition in the "Ca²⁺ before Thap" treatment seems less than in the "Thap before Ca²⁺". Is this consistently observed, and, if so, do the authors have an explanation for this?

25. Figure 7C and legend. A third comment here. Does "ion" refer to the addition of ionomycin? This isn't clear from the description here or in the methods. The change in fluorescence following "ion" addition is less in the Δ TPC traces than in the WT traces. Can the authors comment on why this might be?

26. Page 7 text "in permeabilized cells loaded with the low affinity indicator MagFluo4-AM". The authors could include an explanation of why MagFluo4-AM was used in this experiment (i.e. why it allows them to conclude that there are no defects in Ca²⁺ "handling" in the ER).

Discussion

27. Page 8. "No ion transporters or channels have been reported in the apicoplast until now." The study does not determine whether TgTPC functions an ion channel. Rather, than TgTPC may be important for regulating Ca²⁺ levels in the apicoplast, with its homology to other TPCs consistent with a role as an ion channel.

28. Page 8. "a clear demonstration of the essentiality of the apicoplast metabolic pathways for *T. gondii* motility, host invasion and egress". I disagree that this is a "clear demonstration". "Dead" or non-viable parasites that could result from apicoplast loss would also not be motile or able to invade/egress. Without direct evidence that immediate loss of particular apicoplast pathways results in defects in motility etc, or evidence that parasite are otherwise viable, the authors cannot make this conclusion.

29. Page 9. "similar to that reported for TPC1 at ER-endosome junctions". Could refer to the organism in which this study was performed – "... at ER-endosome junctions in mammalian cells".

30. Page 9. "MCSs" – define abbreviation.

31. Page 9. "identifies TgTPC as an unexpected target for combatting Toxoplasmosis". The authors present no evidence that TgTPC can be targeted by drugs in their study.

Methods

32. Page 10. "a fragment of approximately 2 kbs was amplified from the genomic locus (3' region) of the TgTPC gene using primers 3 and 4 (Table S2)." Primers are in Table S3. There are multiple further instances in the methods where the primer table is incorrectly annotated (as either Table S1 or Table S2).

33. Page 11. The authors describe methods to undertake northern blots, but I don't see these mentioned in the manuscript.

Figures and figure legends not covered in previous comments

34. Figure 1. "HuTPC" vs "HsaTPC" – should be consistent

35. "TPCr" vs "TPCR" – figure abbreviates as TPCR, legend as TPCr. Should be consistent.

36. Figure 2. "500 nM" should read nm

37. Figure 3. "PDH-E" should read PDH-E2

38. Figure 3. "Like the one shown in C" – C depicts electron micrographs, so this doesn't make sense.

39. Figure 5G and 5H. Need to describe the insets from these figures. What are they depicting?

40. Figure 6A. "Anti-TPC" – was this IFA performed with an anti-TPC antibody? Or was anti-cmyc used, as the figure legend text seems to suggest?

41. Figure 6 legend. "dt-Tomato". What does the "dt" represent?

42. Figure 6E legend. "Number of parasites per vacuole" – need to define the time here. i.e. after how many hours of culture. As noted in a previous comment, it would make more sense to describe 6F first.

Reviewer #3 (Remarks to the Author):

The authors have investigated the localization and function of a two-pore channel (TPC) in *Toxoplasma gondii* (Tg). TPCs are found in a wide array of eukaryotic cells, from plants to animals, typically in acidic compartments and vacuoles. The present study identified a novel TPC in Tg, with homologs in many related Apicomplexa, but interestingly not in Plasmodia or related hemosporidians. The Tg TPC is shown here to be localized to the apicoplast, and the authors provide a compelling series of experiments to demonstrate that this protein has a critical role in the maintenance of the functional organelle. Moreover, ablation of the TPC causes a severe replication deficit that greatly reduces intracellular proliferation of the parasite. This phenotype can be overcome by complementation with functional TPC, and importantly, this rescue depends on the presence of specific leucine residues that are critical for its ion channel function. Finally, using targeted genetically-encoded Ca²⁺ indicators, the authors demonstrate that there is uptake of Ca²⁺ released from ER into the apicoplast, and that this depends on the presence of functional TPC.

Overall, this is a well-executed study with an extensive range of conditional knock-in and knock-out Tg mutants that strongly support the primary conclusions regarding the localization and importance of the TPC channel in apicoplast function. The only substantive critique I have is that the title of the manuscript, stating that Ca²⁺ transfer by the TPC is critical for Tg growth, is a step too far. There is relatively little on Ca²⁺ signaling in the paper, and as the authors note themselves, the TPC could be involved in other functions such as forming junctions between the ER and apicoplast. The Ca²⁺ transport role of the TPC could be more strongly supported by manipulating Ca²⁺ levels, and perhaps using known pharmacological effectors of the TPC (which could be validated using the targeted GCaMP6 probes). Nevertheless, the finding that channel pore mutants both interfere with Ca²⁺ transfer to the apicoplast and lead to apicoplast dysfunction, certainly provides enough evidence to propose a Ca²⁺ transfer pathway as the underlying mechanism for the critical role of the Tg TPC. However, without more robust support for the proposed Ca²⁺ transfer function the manuscript title should just be toned down somewhat.

A few other comments and suggestions:

It would help to have some quantification of the immunogold labeling studies.

For Fig. 3A it is not clear why the 3-day panel shows α -ACP and not α -HSP60, especially as the former is presented as red in a merge with red α -HA. It seems that all of the probes are relevant at all time points.

For Fig. 7C, the quantitation should include the magnitude of the ionomycin responses, which seem quite different between RH and Δ -TPC.

The authors should attempt to validate the ability of Tg TPC to allow Ca²⁺ uptake into the apicoplast using a permeabilized cell preparation similar to that used in Fig. S8D. The other panels in this figure show that there is enough signal from the expressed GCaMP6 to record apicoplast Ca²⁺, and such a permeabilized cell experiment would be a good way to show that the leucine mutants block Ca²⁺ permeation.

Most studies of the organellar ion channel functions of the TPC envisage it as a Ca²⁺ release pathway for accumulated Ca²⁺ (eg. from lysosomes). While the authors allude to the known local transfer of Ca²⁺ from ER to mitochondria, the actual driving force for Ca²⁺ uptake through the mitochondrial uniporter is the large membrane potential across the inner mitochondrial membrane. Even with a local Ca²⁺ gradient between ER and apicoplast, established by the leak of Ca²⁺ that occurs in the presence of thapsigargin, is there sufficient driving force to yield the rapid Ca²⁺

uptake seen in Fig. 7C&D?

The discussion should include some thoughts about the potential location and role of a transmembrane ion channel (TPC) in an organelle with four membranes. Also, is it considered that the Ca²⁺ egress pathway is distinct from the proposed TPC uptake pathway, or could the TPC serve both functions? Bearing in mind the title of the paper, a little more discussion of how such a Ca²⁺ signaling pathway could function seems warranted.

Reviewed by Andrew Thomas

REVIEWER

COMMENTS

Reviewer #1 (Remarks to the Author):

The authors have generated a large amount of data to unambiguously demonstrate that TgTPC is involved in maintaining the integrity of the apicoplast and they also bring evidence that the organelle contains Ca²⁺. In the introduction and the discussion, the authors put forward TgTPC as a potential drug target against toxoplasmosis. A number of essential apicoplast proteins have been identified in the past and have yet to be validated as bona fide druggable targets. Not only the manuscript currently provides no clear demonstration as to why TgTPC is essential for apicoplast homeostasis or cell replication (regulating Ca²⁺ availability in the organelle for signalling purposes or providing a co-factor for essential processes?), but the prospect of identifying or designing specific inhibitors of this Ca²⁺ channel may also appear as an unrealistic task as it stands.

We provide new mechanistic data showing that TPC knockout reduces contact site formation between the apicoplast and the ER (Fig. 8). This provides an explanation for why Ca²⁺ transfer between the organelles is compromised in the absence of TPC. Thus, our data suggests that inter-organellar Ca²⁺ signaling by TPCs maintains well-being. Indeed, such signalling is known to be essential for numerous processes across the natural world. It is for this reason, we are of the opinion that TgTPC represents a promising target for chemotherapy. It is essential for parasite replication and our future plans include testing a number of channel blockers against both parasite growth and channel activity. However, as the reviewer indicates, we did not show direct chemical inhibition of TgTPC and we have therefore toned down our claims in the Abstract, the end of the Introduction and the Discussion.

Another main concern I have about this manuscript is the extensive use for phenotypic analysis of a knock-out cell line that has been kept of about a year in culture and has shown striking changes in organelle recovery and growth. I understand the authors took care in providing complemented strains to verify the specificity of the phenotypes, but given the high adaptability of *T. gondii* (and of microorganisms in general) in cultures, it is possible that this cell line is different in many regards from the initial mutant and hence may not be completely adequate to draw clear-cut conclusions.

We now provide whole genome sequence data to address the adaptation question: (NCBI Accession: PRJNA695295) or (<https://www.ncbi.nlm.nih.gov/sra/PRJNA695295>). We find that “The analyzed sequences did not reveal any notable changes in their genome except for a few indels (all within intergenic regions” (lines 203-204). The new data is presented in Table S3. We also include a new figure, Fig S6 showing that the adaptation of the \square TPC mutant is likely the result of the increased levels of apicoplast DNA.

Specific points.

As a general comment, please provide line numbers when formatting the manuscript, it makes it much easier for the reviewers to refer to specific points.

We added line numbers.

Some parts of Fig. 1 (ie conservation of the pore sequence, domain arrangement) are somewhat redundant with the information available in the Prone and Taylor 2011 PloS One paper.

The domain schematic shown in Figure 1B is to scale (unlike Prole and Taylor’s analysis) highlighting the differences in the length of the N-terminus and linker of TgTPC relative to plant and animal TPCs.

The pore alignment in Fig. 1C shows conservation of the pore leucine that we mutated. For these reasons, we would prefer to retain these panels.

It may have been useful to assess for overall solubility of TgTPC, as it would be quite complementary to the localisation at the membrane(s) of the apicoplast.

We agree that this would be a complementary result. Taking into account the large number of new experiments that we had to perform for this response, the limited access to laboratory facilities and the confirmatory nature of this experiment, we rely on the two independent approaches (IFA using superresolution microscopy (Fig. 2E) and electron microscopy (Fig. 2F)) showing that TgTPC localizes to the membrane of the apicoplast. These data are strengthened by additional gold labeling quantification (Line 125-126).

“Somewhat surprisingly, we found little overlap of staining with TgCPL, a marker of the Plant-Like Vacuole (PLV)/VAC””: when discussing TgTPC localization in the corresponding results section, it may be useful to remind the reader of the vacuolar localization of its homolog in higher plants. Also, there is no such thing as ‘data not shown’ these days, please show PLV colocalization as a Suppl. Figure.

We now mention under Results (Lines 117-118) that TPCs normally reside on acidic vacuole. We also include IFA showing the localization of the vacuolar pyrophosphatase and cathepsin L (both PLV markers) with TPC-HA (Fig. S2A-B). These data show no clear overlap although sometimes the organelles appear to be in contact..

pp.4-5. After long-term incubation with ATc, the authors state that “All steps of the lytic cycle therefore were affected TPC in the Apicomplexan parasite *T. gondii* following prolonged-ATc treatment, likely the result of apicoplast fragmentation and its consequent dysfunction.”. Maybe they could be more explicit, as it quite simply seems to reflect a loss of viability.

We now indicate that this is highly reminiscent of the delayed death phenotype previously described (lines 171-174; see below),

Related to that point, on p. 8 they write “Alterations in the apicoplast, as those we report here for the TgTPC conditional mutants, are numerous and a clear demonstration of the essentiality of the apicoplast metabolic pathways for *T. gondii* motility, host invasion and egress”. This is in contradiction with the existing literature. It is commonly accepted that apicoplast loss leads to the so-called ‘delayed death’, by which parasites that have just experienced the loss of the organelle are nonetheless able to egress and invade new hosts, and only upon invading a new host cell they will be unable to replicate without the organelle. In fact, it is strange the term ‘delayed death’ is not even mentioned in this manuscript.

We agree with this comment. Our data agrees with the “delayed death” phenotype because of the mild defect on days 3-5. The severe defect on day 7 is the result of the 2nd or 3rd lytic cycle. The similarity with the delayed-death phenotype is now indicated (Lines 171-174).

p.5. “they stop responding to ATc”. This is confusing as down-regulation still seems effective.

We provide a better explanation for the phenotype (lines 186-188) “were termed.....Tetracycline Resistant because after these passages the small plaques formed by this mutant were not further reduced by ATc”

As mentioned in my general comments, I am wondering if there might be another plastidic ion transporter that would be mutated/upregulated in the adapted TgTPC cell line to perform a similar function as TgTPC, albeit sub-optimally (just enough to recover some apicoplast functions, but not

enough to recover the wild-type phenotypes, which would be only fully restored after complementation with TgTPC). Whole genome sequencing and/or comparative transcriptomic analysis of the initial and the adapted mutant cell line may provide insights into this.

We have now performed whole genome sequencing of the Δ TPC mutant characterized in this work plus that of two additional clones (Δ TPC-*b* and Δ TPC-*cE5*). We also performed whole genome sequencing of the Δ TPC and the Δ TPC-TR mutants. We report the results in Table S3. We did not find any important mutation in other plastid transporter that could explain the recovery of the apicoplast functions (lines 203-204).

We include new data on apicoplast DNA content with time of culture in the new supplemental Figure S6. This data confirms that the mechanism of recovery is likely linked to changes in apicoplast not nuclear DNA. These data together with the sequencing are included in a new Section in the results: “Adaptive changes in TPC null mutants are associated with changes in apicoplast DNA content”.

I am convinced by the Ca^{2+} probe use and measurements. I would just advise the authors to avoid drawing definitive conclusions such as “Thap-induced Ca^{2+} signals in the apicoplast were absent in the Δ TPC-*a* mutants”, as there seems for instance to be a small response on the right graph (when thapsigargin was added after Ca^{2+} load). Also for Fig.7, as several conditions (order for adding Thap and Ca^{2+}) were used, it is not clear to which ones the delta Ca^{2+} and deltaF bar graphs correspond.

We now indicate that “Thap-induced Ca^{2+} signals in the apicoplast were almost absent in the Δ TPC-*a* mutants”. We indicate in the figure legend to which graph the quantification corresponds: “Bar graphs showing the quantification of the Δ Fs after the first addition of Thap (left) or Ca^{2+} (right)”.

Out of curiosity, is it possible to estimate the amounts of Ca^{2+} in the apicoplast? I suppose it may be relatively low as this group previously used elemental mapping and electron microscopy to detect calcium in *T. gondii* tachyzoites and acidocalcisomes seem to represent the main Ca^{2+} store. Related to this, the GCaMP6 showing discernible fluorescence change by microscopic observation? If so, can it be used for live imaging to check if there are Ca^{2+} variations in the apicoplast at specific moment of the cell cycle?

We predict that the concentration of Ca^{2+} in the apicoplast would be similar to the cytosolic concentration under resting conditions and that it would increase after Ca^{2+} uptake following signaling. These signals are difficult to calibrate because of the non-ratiometric nature of the probe. But it should be possible in the future to track dynamic changes at different points in the cell cycle.

Maybe a bit outside the scope of the study, but as there is no TPC homolog in Plasmodium, it would be interesting to express and apicoplast of Ca^{2+} sensor in this Apicomplexa (and it would be quite compelling to find out there is none).

We present a phylogenetic analysis of this gene in Figure 1A. The gene is not present in any hemosporidian. It is only found in coccidian parasites. We have now expanded the phylogenetic analysis and include the Chromerids, the closest known phototrophic relatives to apicomplexan parasites. We discuss in more detail the evolutionary implications of this on lines 85-90.

As the super resolution microscopy data do not bring additional dynamic and kinetic insight into the ER/apicoplast contact sites, it is not more informative than what was already described (and rightfully referred to by the authors in the discussion) by the electron tomography analysis of Tomova et al Traffic 2009. Thus it may be moved to supporting information.

We agree and have moved this data to the supporting information (Figure S11).

NAADP is a known activator of TPC channels for Ca^{2+} flux in animal cells, while this is extensively discussed by the authors, it is frustrating this could not be tested to see if can modulate Ca^{2+} in the *T. gondii* GCaMP6 reporter line. Although NAADP is largely membrane impermeable, several publications show the use of caged compounds (Churchill et al 2002) or even a cell permeant version of NAADP (Parkesh et al Cell calcium 2008), thus I wonder if these may be used in *T. gondii*.

Caged NAADP is no longer commercially available. Cell permeant NAADP (NAADP-AM) is very unstable and difficult to work with. We tested NAADP in digitonin-permeabilized parasites expressing FNR-GCaMP6 but found no response (see figure on the left). This insensitivity might be due to loss of essential accessory proteins.

Reviewer #2 (Remarks to the Author):

1. Figure 5 and associated experiments. The approach to generate a TgTPC knockout in a strain adapted for growth following TgTPC depletion is a novel one. This enables the authors to measure the effects of TgTPC knockout without a concurrent loss of the apicoplast. Presumably, there has been some genetic/epigenetic/metabolic changes that have occurred in these parasites to enable the adaptation to growth in the absence of TgTPC. The authors present no data on what these changes are. Without a clearer explanation of the adaptation that has occurred in these parasites, the subsequent (indirect) experiments analysing the effects of TgTPC on Ca^{2+} changes in the apicoplast are difficult to interpret. Are these changes due to loss of TgTPC, or to adaptations that have occurred in this strain, or (perhaps most likely) a combination of the two? The paper would be strengthened considerably if the authors provided an explanation for what has changed in parasites following long-term culture of the TgTPC knockdown strain. They could perhaps undertake whole genome sequencing to detect adaptive mutations in the genome or RNAseq analysis to measure transcript abundances to better characterize the adaptations that have occurred. I suspect these experiments will take some time to complete and follow up on.

We think that a full mechanistic analysis of the adaptation would be beyond the scope of this study. However, we now include whole genome sequencing of the $\Delta\text{TPC-a}$ mutant plus those of two more clones, the original *i\Delta\text{TPC-TR}* line and the *i\Delta\text{TPC}* conditional mutant (see Table S3 and response to Reviewer 1). We indicate “The analyzed sequences did not reveal any notable changes in their genome except for a few indels (all within intergenic regions” (lines 203-204). The results support our model (speculative for now) that the adaptation occurred through the apicoplast DNA and it is not associated with changes within the nuclear genome. In accord, we present new data in Figure S6 showing that apicoplast DNA content of the mutants increases with time in culture. These data are part of a new section in the Results (lines 198-231).

Please note that the growth defect upon TPC depletion was completely rescued by an extra copy of TPC (Fig. 6D) suggesting strongly that the defect is TPC-dependent. We present new data further indicating that the effects we observed are specific and unrelated to long term adaptation. As shown in Fig. S8D, a conditional copy of TPC also rescues the growth defect upon TPC depletion and growth rescue is reverted when TPC expression is repressed. In other words, the conditional copy in the adapted background

behaves exactly like the original conditional knockout cell line. Any compensatory mechanism should have protected parasites from downregulation of the extra copy of TgTPC. But this was not the case. See lines 243-249.

2. Figure 7. The approach to using Ca²⁺-sensitive GFP fluorescence to examine Ca²⁺ concentrations in the apicoplast is, in principle, a powerful one. As noted in the previous comment, the approach used in generating the TgTPC knockout mutants make it difficult to distinguish between the effects of TgTPC loss and parasite adaptations to TgTPC loss. The authors could consider undertaking control GCaMP6 experiments in both the original Tet-regulatable strain complemented with WT TgTPC and the knockout strain complemented with WT TgTPC to determine whether there are any differences in apicoplast Ca²⁺ responses in parasites expressing TgTPC that have either not adapted (original Tet strain) or adapted (complemented knockout strain) to loss of TgTPC. This could reveal whether adaptation changes overall apicoplast Ca²⁺ responses or not.

We would like to stress that the isolation of lines expressing GCaMP6 in the apicoplast is technically challenging and requires at least 4-6 months. The number of selectable markers required to complement the conditional mutant necessitates selection of GCaMP6 by fluorescence. This is difficult due to the weak signal emanating from a single apicoplast per cell. Additionally, the conditional mutant, when grown with ATc, shows misshaped apicoplasts. It would therefore not be a feasible approach to express a genetic indicator in the organelle during dismembering.

We contend that our new genome sequencing data together with the experiments showing the growth rescue of the mutants by an extra copy of TgTPC and the further growth repression when the extra copy is downregulated answers most of the concerns of this reviewer.

Other comments that require further experimentation:

3. Figure 3C. The apicoplast “appeared swollen”, “fragmentation was confirmed by electron microscopy”. The authors present a single image that claims to show an apparently abnormal apicoplast. This image does not convincingly show either a swollen or a fragmented apicoplast. Without a thorough quantitative analysis of apicoplast morphology upon TgTPC depletion, including measurements of the dimensions of a normal apicoplast and how this changes upon TgTPC depletion, I don’t think the authors can make this claim.

We now include more images as supplemental (Fig. S3C) and we indicate that the apicoplasts appear vacuolated.

4. Figure 4E. Is the defect in ionomycin-dependent egress due to the specific role of TgTPC in apicoplast/parasite biology, or because of apicoplast loss that occurs as a consequence of TgTPC knockdown, and which earlier experiments show is probably substantial after 5 days on ATc? The authors should undertake a more thorough experiment, including (at a minimal) some earlier time points where TgTPC knockdown is substantial (e.g. day 2 post-ATc addition, Fig. 2C), but where some of the morphological defects may be less pronounced (see minor comment 9 below).

This is an interesting observation and brings up an important difference between the conditional and the clean KOs. We do not see an egress phenotype in the *TPC null* cells, so we believe that the egress phenotype that we observed with the conditional mutants is the result of secondary alterations of apicoplast functions. Invasion and egress defects are only seen after 7 and 5 days respectively with ATc

in the conditional mutants. However, replication is significantly different at 4 days with ATc. At this point, the apicoplast labeling is evident but TPC is not expressed.

5. Page 5. “ Δ TPC-a mutants were further cultured for several months, which resulted in gradual recovery of apicoplast labeling with markers like ACP (Fig. 5B).” Without some sort of quantification here, I don’t understand the basis for this statement. Apicoplasts can only arise from pre-existing apicoplasts. Is it that the loss-of-apicoplast phenotype disappears over time?

We do not think that the apicoplast is lost in the conditional mutants. We indicate only decreased labeling (line 193). We contend that the apicoplast is damaged and as a consequence non-functional. However, the genetic material (nuclear and apicoplast DNA) is still present. PCR results showed the presence of apicoplast DNA (lines 205-208). However, at some time points parasites contained very low levels of apicoplast DNA. Our result strongly suggests that the gradual appearance of apicoplast markers is associated with increasing amounts of apicoplast DNA. This is now quantified in the new figure S6. We have isolated 3 knockout clones, and 12 TR clones, all of them were able to restore apicoplast markers after 2-4 months of growth. This is described on lines 216-220. A time course is shown in Fig. S6C (lines 225-229).

6. Figure 6. The authors show that the constitutive expression in the Δ TgTPC strain of mutant versions of TgTPC lacking key Lys residues in the TPC pore fail to rescue growth, and interpret this as evidence that channel activity in TgTPC is required for this protein to carry out its function. An alternative explanation is that the mutant TgTPC proteins are not expressed at sufficient abundance to complement growth (e.g. if the mutation causes destabilization and subsequent degradation of the protein). The authors should perform western blotting to determine how the abundance of the mutant TgTPC isoforms compares to abundance of the complementing WT TgTPC protein.

Please note that the IFA in Figure 6A showed comparable signals from the wildtype and mutant lines. Nevertheless, westerns blots of the three cell lines are now presented in Figure S8B showing similar expression. This is described on lines 274-275.

7. Figure 7D-E. The apparent interaction between the apicoplast and ER is intriguing, and something that has been noted in previous studies. However, the authors conclusions are based on only three images. If they want to include these data, they should quantify the extent to which the ER and apicoplast interact.

We include a new Figure 8 in which we quantified interaction by assessing the proportion of apoplasts in contact with the ER and the distance between them in 50 randomly chosen cells. Intriguingly, comparison with the Δ TPC mutant shows that contact is reduced upon TPC depletion (Fig. 8B) thereby providing a possible mechanism underpinning reduced Ca^{2+} transfer This data is described in lines 337-345 and discussed in lines 393-396,

Minor comments:

Introduction

1. Page 2. “But TPCs are also activated by the endosomal phosphoinositide”. The phrasing is confusing here. Are all TPCs activated by PI(3,5)P₂? Or only a subset (like TPC1 and 2 from mammals)?

A subset. This is now clarified: “But some TPCs may also be activated by the endosomal phosphoinositide, PI(3,5)P₂” (line 54).

2. Page 2. Is it correct to refer to the apicoplast as a “chloroplast remnant”? It is a plastid that retains numerous canonical plastid functions, so is not really a “remnant” organelle.

Changed it to non-photosynthetic plastid (line 66).

3. Page 2. What is the evidence that the vacuole of these organisms is plant-derived in an evolutionary sense? Many other organisms have vacuoles or lysosomes, which appear to be functionally similar to the describe PLV/VAC compartment.

We did not state it is plant-derived but plant-like (line 67).

Results and figures

4. Figure 1A. Given the discussion of plant TPCs in the introduction and the link of TgTPC with the apicoplast, the phylogenetic tree would benefit from inclusion of some plant TPCs. Are the sequences of plant TPCs conserved to a sufficient extent to enable them to be included in the phylogenetic analysis?

Plant sequences are now included.

5. Figure 1A. The tree is missing representatives of the closest free-living relatives to apicomplexans (e.g. chromerids). Do the TPC homologues from chromerids group with the coccidian TPC clade (which would suggest a homologue was present in the common free-living ancestor of apicomplexans and lost in hemosporidians) or do coccidian TPCs form a distinct clade as argued in the text? I think an expanded analysis would add more value to the evolutionary angle of the study.

Two chromerid sequences are now included in a completely revised tree (Fig. 1A). These form a group with the apicomplexan sequences distinct from known TPCs. Absence of TPCs in hemosporidians is thus likely due to lineage-specific loss of TPCs. This is discussed briefly in lines 85-90.

6. Figure S1. I found the Southern blots difficult to interpret. What are the expected sizes of the bands on this Southern blot upon locus modification to introduce the 3HA and the tetracycline-regulatable element? And do these match to the observed bands? The 3.9 kb band expected for probe 2 in WT parasites in the diagram is the same size as the band observed in the modified locus with probe 1. This raises concerns about the strategy to use probe 2 on a membrane where the signal from probe 1 is still visible. The authors should provide more details on how they verified the genetic modifications.

Corrected. The size is different. The 3.9kb arrow should have pointed to the lower band.

7. Page 3. “we found little overlap of staining with TgCPL ... (not shown)”. Given that TPCs localize to vacuoles/lysosomes in other organisms, perhaps worth showing these data in a supplementary figure?

These data are now shown in supplemental Figure S2A-B.

8. Figure 2E. This image is very zoomed-in. This makes the context of the labelled structures within the parasite cell difficult to establish. The authors should include an image of the entire cell here.

Additional expanded images are now shown as supplemental Figure S2C.

9. Figure S2. The data demonstrate that defects in apicoplast biogenesis are apparent 3 days after ATc addition. Later experiments in the study examine processes such as egress at time points that occur well after these biogenesis defects are apparent. The paper would be strengthened by establishing whether apicoplast loss is concomitant with TgTPC depletion, or whether apicoplast loss occurs subsequent to TgTPC knockdown. This could be achieved by including earlier time points (e.g. 1 and 2 days) in this experiment.

According to our results, TgTPC depletion occurs before the apicoplast marker decreased detection. See Fig 3A where TPC expression is not detectable after 3 days of down regulation but apicoplast markers are still evident. Only at 7 days +ATc the apicoplast markers become undetectable.

10. Figure 3A and page 4 “the apicoplast marker ACP ... was no longer detectable after culturing Δ TPC-3HA with ATc for 7 days.” According to the figure, the day 7 sample was labelled with anti-Hsp60 as the apicoplast marker not anti-ACP.

Corrected. We used anti-hsp60 (lines 134-135).

11. Figure S3 legend. Need to describe what the errors bars represent in these column graphs and what statistical analysis was performed.

Data are presented as mean \pm SD and statistical analysis was done with the Student's t-test using GraphPad Prism. This is now included in the legend.

12. Figure 3D. The antibody used is not clear. The text in the main body states that it is an anti-lipoic acid antibody, whereas the figure legend states anti-lipoylated PDH-E2. The authors should clarify this. The methods suggests an anti-lipoylated PDH-E2 antibody was used, in which case the main body of the text should be changed.

We used anti-lipoylated PDH-E2 antibody. Text in the main body has been corrected (line 151).
Apologies for the confusion

13. Figure 3D, column graph. The authors should indicate what the error bars depict, and what statistical analysis was performed.

We included this information in the figure legend (SD and Student's *t*-test)

14. Figure 4A-B. Are the growth defects due solely to depletion of TgTPC? These data would be strengthened by inclusion of a strain wherein a constitutively-expressed TgTPC is added back to the regulatable strain to see if this complements parasite growth upon depletion of the regulated TgTPC gene (as the authors do in later experiments in the tetracycline-resistant strain).

Our data indicate that the growth defects seen with the conditional depletion are due to a combination of defects caused by the depletion of TgTPC plus additional defects due to apicoplast dysfunction. We decided not to complement the conditional mutant because we focused the work on the function of the TPC. This was the main reason we designed the strategy shown in Figure S4 to obtain clean KOs so we could study the function of TPC alone. However, we created two different complementations of the clean KOs that are described in detail in Figs 5, S7 and S8.

15. Figure 4B-F. Need to indicate what the error bars represent.

Added (SD and Student's *t*-test).

16. Figure S3B and text. The Tet-resistant strains “expressed neither TgTPC (Fig. S3B) nor ACP (Fig. S3B)”. Western blotting is more appropriate than IFAs of measuring whether these proteins are expressed or not, and it would seem unlikely that ACP is not expressed at all in these parasites. Perhaps the more general point to make is that apicoplast appears to be absent in the resistant strain.

This is now Figure S4. We agree and we state that the apicoplast appears to be absent in the resistant strain (Lines 187-188).

17. Figure 5D, F-H. Need to describe what error bars in graphs represent. Where relevant, need to describe statistical analysis that was performed.

Added (SD and Student's *t*-test).

18. Page 6 text. “(Fig. 6B middle panels and B)”. Remove the “and B”? Do the authors mean to refer to the right image of the middle panel? This needs clarification.

Corrected and clarified “(Fig. 6B middle panels, $\Delta TPCa$ and $\Delta TPC-a-iTPC$)”.

19. Figure S7C. Figure legend needs to describe what the error bars represent. I've noted this elsewhere as well, but in all graphs depicted in the manuscript, the authors must describe what error bars represent (almost always missing), and present a description of the statistics that they carried out (occasionally missing).

All relevant Figure legends have been amended to state values are means \pm SD, and use of Student's *t*-test. We have also included a paragraph on Statistics in the Materials and Methods (line 584-586)

20. Figure 6E-F. Given that 6E represents the averaged data from 6F, it makes more sense to show these the other way round, or combined into a single figure.

We switched them as suggested and now figure 6E represents the % of PVs with 1,2,4,8, and 16 parasites while Fig. 6F shows the mean number of parasites per vacuole at 24 h.

21. Page 6. “these data indicate that the TgTPC directly impacts *T. gondii* replication and the pore domain of TgTPC is essential for its function”. Since they don't have an assay that directly measures TgTPC pore activity (e.g. electrophysiology studies on purified apicoplasts or on TgTPC expressed in a heterologous system), this conclusion is over-reaching what the data show. Rather, the authors should inject a note of caution “... data are consistent with the pore domain of TgTPC being important for its function.”

Changed to “data are consistent with...”.(line 290)

22. Figure 7. Why was RH used as a WT strain here and not TATi Δ ku80 (i.e. the parental strain)?

We used RH because we wanted to compare the mutant with the parental cell line for which we have the most information on Ca²⁺ homeostasis and signaling and also with no mutations. We did not see major differences in the cytoplasmic Ca²⁺ between RH and the mutant which was encouraging because it meant that any unrelated mutations of the line TATi Δ ku80 did not impact the result.

23. Figure 7C. The authors need to specify what the $\Delta F/F_0$ value on the y axes represent. The changes in apicoplast [Ca²⁺] upon thapsigargin addition in WT parasites are small. Although the authors find statistical significance in the data (ΔF after Thap in WT vs mutant parasites), I think it would help readers if the variation in the data were better represented. Ideally, the authors would plot $\Delta F/F_0$ data points from each individual experiment on the graph rather than representing the data as column graphs. At the very least the authors need to specify what the error bars represent (SD, CI or SEM?).

$\Delta F/F_0$ represents the change in fluorescence relative to the basal fluorescence. This is now stated in the Legend.

Individual data points are now added to the bar graphs. Error bars represent SD. This is now stated in the Legend.

24. Figure 7C legend. As a second comment on Figure 7C, do the values on the column graphs to the right of the traces represent the ΔF after thapsigargin in the “Thap before Ca²⁺” traces, or a combination of the “Thap before Ca²⁺” and “Ca²⁺ before Thap” traces? The increase in fluorescence following thapsigargin addition in the “Ca²⁺ before Thap” treatment seems less than in the “Thap before Ca²⁺”. Is this consistently observed, and, if so, do the authors have an explanation for this?

The quantification presented in Fig 7C corresponds to the ΔF after thapsigargin in the “Thap before Ca²⁺” traces (left ones). This is now stated in the Legend. The increase in fluorescence following thapsigargin addition in the two conditions was not significantly different.

25. Figure 7C and legend. A third comment here. Does “ion” refer to the addition of ionomycin? This isn’t clear from the description here or in the methods. The change in fluorescence following “ion” addition is less in the ΔTPC traces than in the WT traces. Can the authors comment on why this might be?

Yes. Ion is ionomycin. This is now stated in the Legend. The ionomycin responses are now quantified in Fig. 7F.

26. Page 7 text “in permeabilized cells loaded with the low affinity indicator MagFluo4-AM”. The authors could include an explanation of why MagFluo4-AM was used in this experiment (i.e. why it allows them to conclude that there are no defects in Ca²⁺ “handling” in the ER).

MagFluo4 is a low affinity calcium indicator that compartmentalizes into intracellular organelles when the parasites are incubated for extended periods with high concentrations of its AM ester. It thus reports high luminal Ca²⁺ levels typically in the ER (due to its relatively large volume) upon cell permeabilization. Ca²⁺ uptake following addition of Ca²⁺ in the presence of MgATP is a measure of the SERCA activity, which is almost identical between RH and the ΔTPC mutant.

An explanation and a reference (38) are now included in the main body of the text (line 325) and in the Legend to Fig. S10F.

Discussion

27. Page 8. “No ion transporters or channels have been reported in the apicoplast until now.” The study does not determine whether TgTPC functions an ion channel. Rather, than TgTPC may be important for regulating Ca²⁺ levels in the apicoplast, with its homology to other TPCs consistent with a role as an ion channel.

The wording has been changed in the discussion to “No regulators of Ca²⁺ levels in the apicoplast have been reported until now”. Line 368-369

28. Page 8. “a clear demonstration of the essentiality of the apicoplast metabolic pathways for T. gondii motility, host invasion and egress”. I disagree that this is a “clear demonstration”. “Dead” or non-viable parasites that could result from apicoplast loss would also not be motile or able to invade/egress. Without direct evidence that immediate loss of particular apicoplast pathways results in defects in motility etc, or evidence that parasite are otherwise viable, the authors cannot make this conclusion.

The sentence was deleted.

29. Page 9. “similar to that reported for TPC1 at ER-endosome junctions”. Could refer to the organism in which this study was performed – “... at ER-endosome junctions in mammalian cells”.

Amended as suggested (line 396).

30. Page 9. “MCSs” – define abbreviation.

Abbreviation (meaning membrane contact site) was deleted as it was used only once.

31. Page 9. “identifies TgTPC as an unexpected target for combatting Toxoplasmosis”. The authors present no evidence that TgTPC can be targeted by drugs in their study.

The sentence was modified to “potential target” (line 402).

Methods

32. Page 10. “a fragment of approximately 2 kbs was amplified from the genomic locus (3’ region) of the TgTPC gene using primers 3 and 4 (Table S2).” Primers are in Table S3. There are multiple further instances in the methods where the primer table is incorrectly annotated (as either Table S1 or Table S2).

Apologies. This has been corrected throughout. Primers are now in Table S4.

33. Page 11. The authors describe methods to undertake northern blots, but I don’t see these mentioned in the manuscript.

Apologies. This has been deleted.

Figures and figure legends not covered in previous comments

34. Figure 1. “HuTPC” vs “HsaTPC” – should be consistent

We have changed the nomenclature to Hs to better match Tg.

35. “TPCr” vs “TPCR” – figure abbreviates as TPCR, legend as TPCr. Should be consistent.

Changed to TPCR in the Legend.

36. Figure 2. “500 nM” should read nm

Corrected.

37. Figure 3. “PDH-E” should read PDH-E2

Corrected to PDH-E2.

38. Figure 3. “Like the one shown in C” – C depicts electron micrographs, so this doesn’t make sense.

Deleted.

39. Figure 5G and 5H. Need to describe the insets from these figures. What are they depicting?

Inset in Fig. 5G shows the quantification of three independent experiments at day 8. Inset in Fig. 5H shows cell numbers on day 8 from 3 independent experiments. This is now described in the legend.

40. Figure 6A. “Anti-TPC” – was this IFA performed with an anti-TPC antibody? Or was anti-cmyc used, as the figure legend text seems to suggest?

It was anti-c-Myc. Corrected.

41. Figure 6 legend. “dt-Tomato”. What does the “dt” represent?

td refers to Tandem dimer. This was corrected and defined in the text (Line 497).

42. Figure 6E legend. “Number of parasites per vacuole” – need to define the time here. i.e. after how many hours of culture. As noted in a previous comment, it would make more sense to describe 6F first.

24 hours, included in the legend. Figs. E and F are switched as suggested

Reviewer #3 (Remarks to the Author):

Overall, this is a well-executed study with an extensive range of conditional knock-in and knock-out Tg mutants that strongly support the primary conclusions regarding the localization and importance of the TPC channel in apicoplast function.

The only substantive critique I have is that the title of the manuscript, stating that Ca²⁺ transfer by the TPC is critical for Tg growth, is a step too far. There is relatively little on Ca²⁺ signaling in the paper, and as the authors note themselves, the TPC could be involved in other functions such as forming junctions between the ER and apicoplast. The Ca²⁺ transport role of the TPC could be more strongly supported by manipulating Ca²⁺ levels, and perhaps using known pharmacological effectors of the TPC (which could be validated using the targeted GCaMP6 probes). Nevertheless, the finding that channel pore mutants both interfere with Ca²⁺ transfer to the apicoplast and lead to apicoplast dysfunction, certainly provides enough evidence.

We have changed the title to “A plastid Two-Pore Channel essential for inter-organelle communication and growth of *Toxoplasma gondii*”,

A few other comments and suggestions:

It would help to have some quantification of the immunogold labeling studies.

We quantified gold marks comparing apicoplast membrane with lumen and with other cellular compartments (lines 125-127).

For Fig. 3A it is not clear why the 3-day panel shows α -ACP and not α -HSP60, especially as the former is presented as red in a merge with red α -HA. It seems that all of the probes are relevant at all time points.

We corrected the α -ACP color. We wanted to use two different apicoplast markers to demonstrate that it was a general apicoplast defect.

For Fig. 7C, the quantitation should include the magnitude of the ionomycin responses, which seem quite different between RH and Δ -TPC.

We include now quantification of the response to Ionomycin as Figure 7F.

The authors should attempt to validate the ability of Tg TPC to allow Ca^{2+} uptake into the apicoplast using a permeabilized cell preparation similar to that used in Fig. S8D. The other panels in this figure show that there is enough signal from the expressed GaCMP6 to record apicoplast Ca^{2+} , and such a permeabilized cell experiment would be a good way to show that the leucine mutants block Ca^{2+} permeation.

We did this and we saw no uptake. See figure with two sequential Ca^{2+} additions: 220 nM and 375 μM free Ca^{2+} .

Most studies of the organellar ion channel functions of the TPC envisage it as a Ca^{2+} release pathway for accumulated Ca^{2+} (eg. from lysosomes). While the authors allude to the known local transfer of Ca^{2+} from ER to mitochondria, the actual driving force for Ca^{2+} uptake through the mitochondrial uniporter is the large membrane potential across the inner mitochondrial membrane. Even with a local Ca^{2+} gradient between ER and apicoplast, established by the leak of Ca^{2+} that occurs in the presence of thapsigargin, is there sufficient driving force to yield the rapid Ca^{2+} uptake seen in Fig. 7C&D?

This is a great question for which we do not have an answer yet. Experiments with intact parasites showed that Ca^{2+} moves from the ER to apicoplast with thapsigargin addition. To confirm that the uptake of Ca^{2+} by the apicoplast stimulated with Thap derives from the ER, we used permeabilized parasites and found that Thap addition resulted in increase of the GCaMP6 fluorescence which was significantly greater in the

parental compared to the Δ TPC strain (Figure A). This increase was dependent on MgATP (Figure B). These data suggest direct transfer from the ER because we did not see Ca^{2+} uptake by the apicoplast by adding high Ca^{2+} alone to the buffer (see reponse to previous point). We are currently investigating our ‘tunnelling’ hypothesis

The discussion should include some thoughts about the potential location and role of a transmembrane ion channel (TPC) in an organelle with four membranes. Also, is it considered that the Ca^{2+} egress pathway is distinct from the proposed TPC uptake pathway, or could the TPC serve both functions? Bearing in mind the title of the paper, a little more discussion of how such a Ca^{2+} signaling pathway could function seems warranted.

We now discuss these points in lines 396-400.

Reviewed by Andrew Thomas

Reviewer #1 (Remarks to the Author):

To me the adaptive changes of the TPC mutants remain enigmatic. However, I praise the considerable efforts made by the authors to reply to all the points that were raised and I have no further comment on this manuscript.

Reviewer #2 (Remarks to the Author):

The authors present evidence that TgTPC is important for proliferation of *T. gondii*, with a specific defect in apicoplast biology. They demonstrate that loss of TPC is correlated with defects in Ca²⁺ uptake by the apicoplast, and also correlate TPC loss with impaired connections between the apicoplast and ER. They have addressed most of my comments and criticisms on the first iteration of the manuscript (and I particularly appreciate their quantification of the interactions between the apicoplast and ER). Overall, the authors present a very interesting story and (aside from the criticisms that I outline below) have produced an excellent body of work.

Despite genome sequencing, the authors do not provide insights into what enables the Δ TPC parasites to adapt to loss of TPC. The authors state that doing so is beyond the scope of the manuscript. I agree that, with the focus of the manuscript on the function of TPC, this gets away from the main story that they present. But I also think that this limits the strength of the conclusions they can make on the role of TPC in apicoplast Ca²⁺ uptake and apicoplast/ER connections (see next comment). They present solid evidence via the complementation experiments that the growth phenotypes they observe are primarily the result of TPC loss. But see my major comment 2 below – I disagree with their conclusion that the adaptation is definitively not due to compensatory effects.

They also don't address my second major criticism on the first draft (see major comments 3 and 4 below). Their Ca²⁺ indicator experiments show a difference in Ca²⁺ responses in the apicoplast between WT and Δ TPC parasites. However, given the compensatory effects that have occurred to generate the Δ TPC strain (for example, perhaps a compensatory effect that disconnects the ER from the apicoplast shown in Figure 8), they cannot be certain that the effect they observe is because of (or solely because of) TPC loss. As noted in my initial comments, the data would be strengthened if the authors tested whether complementation of the Δ TPC strain with WT TPC could restore the Ca²⁺ phenotype (and apicoplast-ER connection phenotype) that they observe.

Major comments

1. Line 191. "Southern blots (Fig S5A) confirmed deletion of the TgTPC gene." This still isn't clear to me from the data shown. The Southern blot data appear the same as the i Δ TPC strain in Figure S1, from which the Δ TPC-a strain is derived. The authors should clarify this (and include additional data that demonstrate knockout of TPC if the current data are ambiguous). Perhaps including a schematic showing the expected sizes of the fragment(s), and how this distinguishes Δ TPC from the parental strain, would help.

2. Line 247-248. "This result confirmed that the adaptation of the Δ TPC-a mutant (and the other genome sequenced clones) is not due to compensatory mutations in its nuclear genome." While this is possible, I don't think the data are confirmatory as the authors claim (and something must account for the compensation). The knockout mutant grows better than the non-adapted knockdown mutant (compare the Δ TPC-a condition in Figure 5F to the i Δ TPC+ATc condition Figure 4B; and compare the Δ TPC-a condition to the i Δ TPC+ATc in Figure S8D) but not as well as WT or i Δ TPC -ATc or Δ TPC-a-iTPC -ATc parasites. Complementation the knockout mutant with TPC restores growth to approximately WT levels, indicating that the most of the growth phenotype is due to TPC loss, but other (compensatory) effects must also have occurred to account for the differences

between the phenotypes in the “adapted” Δ TPC-a strain and the +ATc condition in the i Δ TPC strain (whether this is due to mutations in the nuclear genome or epigenetic or other adaptive effects). Given the challenges associated with characterising exactly what has happened in the adapted strain, the authors could address this point by toning down their conclusions here.

3. Figure 7D-E and associated discussion. As outlined above, given the compensatory effects that have occurred in the Δ TPC strain, I don’t think the authors can state definitively that the defects in [Ca²⁺] changes that they observed in the Δ TPC-a strain are solely due to loss of TPC. They have generated a complemented line (Δ TPC-a-iTPC) – does complementation restore the observed increase in [Ca²⁺] upon thapsigargin addition?

4. Figure 8 and associated discussion (e.g. lines 346-7). The idea that TPC is regulating the association of the ER and apicoplast is an intriguing one, but can the authors rule out that the compensatory mutations that they observe in generating the Δ TPC strain are what is causing this dissociation? They could test this by measuring whether contact sites are restored in complemented TPC mutants.

Other Comments

5. Line 62. “the protist kingdom”. Protists are paraphyletic, so from an evolutionary viewpoint, it doesn’t make sense to consider them a kingdom.

6. Figure 1 legend. “sequences from coccidians and other select unicellular organisms and metazoans ” ... and also plants.

7. Figure S1. I find it difficult to make sense of this figure when only the native locus is depicted in the schematic. Consider including schematics of the various modified strains and the expected sizes of the digested fragments on each.

8. Figure 2 legend. “The HA signal co-localizes with ... the DAPI signal characteristic of the apicoplast”. Without a separate image of the DAPI label, this isn’t clearly visible on the images presented.

9. The left panel in Figure 2F and the top right panel in Figure S2 are duplicated.

10. Line 149. “lipoylation” not “lypoylation”

11. Line 150-153. “Western blot analysis with anti-lipoylated PDH-E2 antibody (Fig. 3D) revealed that the apicoplast lipoylation pathways was reduced at days 3 and 5 compared to controls in the i Δ TPC-3HA(+ATc) and was absent after 7 days with ATc (Fig. 3D).” The authors cannot rule out that a depletion of the PDH-E2 protein, rather than a reduction in the “lipoylation pathways” is responsible for the observed decrease in abundance. Do other proteins (Hsp60, ACP) decrease in abundance upon TPC knockdown? If not, this may lend some credence to the description of lipoylation being reduced. Otherwise, the authors need to consider the possibility that apicoplast proteins are more generally depleted in their interpretation.

12. Line 202. “sequenced” not “equenced”

13. Figure S6 legend. This was cut off in the pdf of the manuscript that I read.

14. Line 230-231. “these data show that apicoplast recovery in TPC-depleted cells is likely driven by increases in apicoplast DNA”. How can the authors distinguish between causation (as they imply) or correlation (i.e. that some factor that enables Δ TPC parasites to grow and successfully replicate their apicoplast also results in an increase in apicoplast DNA? I think they should tone down this conclusion.

15. Line 231. “Portion” – incomplete sentence?

16. Figure 5F legend. Need to describe what error bars represent.

17. Figure S8B. What antibodies are being used in these two panels? c-Myc top and tubulin bottom? This isn't clear from the description in the figure panel or legend.
18. Line 252-3. "motility was only modestly affected upon TPC knockout (Figs. S9C and D)" – should refer only do S9D, since S9C shows the iΔTPC strain (or just cite both figures at the end of this sentence).
19. Line 253-4. "These differences indicate that the slow growth phenotype of the ΔTPC-a mutant results specifically from a replication defect." This is certainly the most likely explanation, but the authors don't actually measure this until Fig. 6E. Perhaps rephrase or make this conclusion later?
20. Figure 6C and legend. What is the "ΔTPC-cm" condition described in the figure? This isn't described elsewhere. Presumably the ΔTPC-a-iTPC strain mentioned in the legend? The authors should make this clearer.
21. Figure 7B. Figure 7D shows the effects of ionomycin addition on apicoplast [Ca²⁺], but the equivalent data are not shown for cytosolic [Ca²⁺]. The authors mention that the traces were obtained simultaneously, so why not show this effect?
22. Line 338, "delta symbol" rather than D in "TatiΔku80"
23. Figure 8C. The authors should clarify what the arrows depict.
24. Line 400. "lumen" rather than "matrix"?
25. Lines 400-402. "Regardless of the exact mechanism, communication between the apicoplast and the ER is essential for apicoplast function and cell growth and identifies". As noted in my major comment 4 above, alternatively, it is the compensatory loss of connection between the ER and apicoplast that enables ΔTPC parasites to proliferate. Without data that the observed ER/apicoplast connections are due to loss of TPC and not compensatory mutations that have occurred in this strain, the authors should temper their conclusions.
26. Line 487. Why "plant and animal" whole genome re-sequencing?
27. Line 574. In the description of the TEM studies, were the authors blinded to whether they were observing WT or ΔTPC parasites?

Reviewer #3 (Remarks to the Author):

The revised manuscript has addressed all of the points raised in my previous review.

Andrew Thomas

REVIEWER COMMENTS

Reviewer #1 (Remarks to the Author):

To me the adaptive changes of the TPC mutants remain enigmatic. However, I praise the considerable efforts made by the authors to reply to all the points that were raised and I have no further comment on this manuscript.

We thank the reviewer for appreciating the considerable effort we have made in revising the manuscript.

Reviewer #2 (Remarks to the Author):

The authors present evidence that TgTPC is important for proliferation of *T. gondii*, with a specific defect in apicoplast biology. They demonstrate that loss of TPC is correlated with defects in Ca²⁺ uptake by the apicoplast, and also correlate TPC loss with impaired connections between the apicoplast and ER. They have addressed most of my comments and criticisms on the first iteration of the manuscript (and I particularly appreciate their quantification of the interactions between the apicoplast and ER). Overall, the authors present a very interesting story and (aside from the criticisms that I outline below) have produced an excellent body of work.

We are pleased that the reviewer considers this an interesting and excellent body of work.

Despite genome sequencing, the authors do not provide insights into what enables the Δ TPC parasites to adapt to loss of TPC. The authors state that doing so is beyond the scope of the manuscript. I agree that, with the focus of the manuscript on the function of TPC, this gets away from the main story that they present. But I also think that this limits the strength of the conclusions they can make on the role of TPC in apicoplast Ca²⁺ uptake and apicoplast/ER connections (see next comment). They present solid evidence via the complementation experiments that the growth phenotypes they observe are primarily the result of TPC loss. But see my major comment 2 below – I disagree with their conclusion that the adaptation is definitively not due to compensatory effects.

We are pleased that reviewer agrees that delineating the mechanism of compensation is beyond the scope of the study.

They also don't address my second major criticism on the first draft (see major comments 3 and 4 below). Their Ca²⁺ indicator experiments show a difference in Ca²⁺ responses in the apicoplast between WT and Δ TPC parasites. However, given the compensatory effects that have occurred to generate the Δ TPC strain (for example, perhaps a compensatory effect that disconnects the ER from the apicoplast shown in

Figure 8), they cannot be certain that the effect they observe is because of (or solely because of) TPC loss. As noted in my initial comments, the data would be strengthened if the authors tested whether complementation of the Δ TPC strain with WT TPC could restore the Ca²⁺ phenotype (and apicoplast-ER connection phenotype) that they observe.

We have succeeded in performing the technically demanding complementation experiments.

Major comments

1. Line 191. “Southern blots (Fig S5A) confirmed deletion of the TgTPC gene.” This still isn’t clear to me from the data shown. The Southern blot data appear the same as the $i\Delta$ TPC strain in Figure S1, from which the Δ TPC-a strain is derived. The authors should clarify this (and include additional data that demonstrate knockout of TPC if the current data are ambiguous). Perhaps including a schematic showing the expected sizes of the fragment(s), and how this distinguishes Δ TPC from the parental strain, would help.

We repeated the Southern blot several times and combined the result for publication. We have now included an additional schematic in Figure S1 and modified Fig S5A. For the demonstration of the Δ TPC KO, the membrane was first hybridized to a probe against the 5’-ORF of TPC and the result shows clearly that there is no signal from the Δ TPC-a cells. The membrane was subsequently hybridized to the probe against the 3’-UTR of TPC. The signal was unchanged and confirmed equal loading of digested genomic DNA. Therefore, we concluded that the lack of TPC 5’-ORF signal is because the TgTPC gene was knocked out. Knockout was confirmed by sequencing (see below).

2. Line 247-248. “This result confirmed that the adaptation of the Δ TPC-a mutant (and the other genome sequenced clones) is not due to compensatory mutations in its nuclear genome.” While this is possible, I don’t think the data are confirmatory as the authors claim (and something must account for the compensation). The knockout mutant grows better than the non-adapted knockdown mutant (compare the Δ TPC-a condition in Figure 5F to the $i\Delta$ TPC+ATc condition Figure 4B; and compare the Δ TPC-a condition to the $i\Delta$ TPC+ATc in Figure S8D) but not as well as WT or $i\Delta$ TPC -ATc or Δ TCP-a- i TPC -ATc parasites. Complementation of the knockout mutant with TPC restores growth to approximately WT levels, indicating that the most of the growth phenotype is due to TPC loss, but other (compensatory) effects must also have occurred to account for the differences between the phenotypes in the “adapted” Δ TPC-a strain and the

+ATc condition in the i Δ TPC strain (whether this is due to mutations in the nuclear genome or epigenetic or other adaptive effects). Given the challenges associated with characterising exactly what has happened in the adapted strain, the authors could address this point by toning down their conclusions here.

We toned down the conclusion, lines 248-252

3. Figure 7D-E and associated discussion. As outlined above, given the compensatory effects that have occurred in the Δ TPC strain, I don't think the authors can state definitively that the defects in [Ca²⁺] changes that they observed in the Δ TPC-a strain are solely due to loss of TPC. They have generated a complemented line (Δ TPC-a-iTPC) – does complementation restore the observed increase in [Ca²⁺] upon thapsigargin addition?

We have now complemented the Δ TPC-a-FNR-GCaMP6-HA mutant expressing FNRGCaMP6 with the TgTPC gene and performed rescue experiments. The results show 100% recovery of apicoplast Ca²⁺ uptake in the mutants expressing the complemented copy of TPC (Fig. 7e-g). These data are described in Lines 330-333 and the method is described in lines 579-583.

4. Figure 8 and associated discussion (e.g. lines 346-7). The idea that TPC is regulating the association of the ER and apicoplast is an intriguing one, but can the authors rule out that the compensatory mutations that they observe in generating the Δ TPC strain are what is causing this dissociation? They could test this by measuring whether contact sites are restored in complemented TPC mutants.

We now include additional EM experiments in Figure 8 with the Δ TPC-TPC complemented mutant. The results show 100% recovery of contact sites. Line 353.

Other Comments

5. Line 62. “the protist kingdom”. Protists are paraphyletic, so from an evolutionary viewpoint, it doesn't make sense to consider them a kingdom.

Corrected.

6. Figure 1 legend. “sequences from coccidians and other select unicellular organisms and metazoans ” ... and also plants.

Corrected.

7. Figure S1. I find it difficult to make sense of this figure when only the native locus is depicted in the schematic. Consider including schematics of the various modified strains and the expected sizes of the digested fragments on each.

We modified Figure S1 as the reviewer suggested and include a second cartoon showing the expected fragments in the KOs.

8. Figure 2 legend. “The HA signal co-localizes with ... the DAPI signal characteristic of the apicoplast”. Without a separate image of the DAPI label, this isn’t clearly visible on the images presented.

Figure 2D, HA panel shows the colocalization of a-HA signal (red) with DAPI signal (blue). Especially with the Δ TPC cells, it is possible to see the ring shape signal of a-HA surrounding the DAPI signal. We now include a white arrow pointing at the blue signal surrounded by the red signal.

9. The left panel in Figure 2F and the top right panel in Figure S2 are duplicated.

That is correct but we included Fig. S2 because the reviewer asked for an enlarged version of 2F.

10. Line 149. “lipoylation” not “lypoylation”

Corrected.

11. Line 150-153. “Western blot analysis with anti-lipoylated PDH-E2 antibody (Fig. 3D) revealed that the apicoplast lipoylation pathways was reduced at days 3 and 5 compared to controls in the Δ TPC-3HA(+ATc) and was absent after 7 days with ATc (Fig. 3D).” The authors cannot rule out that a depletion of the PDH-E2 protein, rather than a reduction in the “lipoylation pathways” is responsible for the observed decrease in abundance. Do other proteins (Hsp60, ACP) decrease in abundance upon TPC knockdown? If not, this may lend some credence to the description of lipoylation being reduced. Otherwise, the authors need to consider the possibility that apicoplast proteins are more generally depleted in their interpretation.

The sentence was re-phrased, line 154-155

12. Line 202. “sequenced” not “equenced”

Corrected.

13. Figure S6 legend. This was cut off in the pdf of the manuscript that I read.

Corrected.

14. Line 230-231. “these data show that apicoplast recovery in TPC-depleted cells is likely driven by increases in apicoplast DNA”. How can the authors distinguish between causation (as they imply) or correlation (i.e. that some factor that enables Δ TPC

parasites to grow and successfully replicate their apicoplast also results in an increase in apicoplast DNA? I think they should tone down this conclusion.

We have toned this down as follows: 'Taken together, these data show that the recuperation of the apicoplast DNA impacted the growth of TPC-depleted cells.' Line 232-233

15. Line 231. "Portion" – incomplete sentence?

Corrected.

16. Figure 5F legend. Need to describe what error bars represent.

Corrected in the legend.

17. Figure S8B. What antibodies are being used in these two panels? c-Myc top and tubulin bottom? This isn't clear from the description in the figure panel or legend.

Corrected. Added in the legend that the loading control shows the tubulin signal

18. Line 252-3. "motility was only modestly affected upon TPC knockout (Figs. S9C and D)" – should refer only do S9D, since S9C shows the Δ TPC strain (or just cite both figures at the end of this sentence).

Corrected. Cited at the end of the sentence.

19. Line 253-4. "These differences indicate that the slow growth phenotype of the Δ TPC-a mutant results specifically from a replication defect." This is certainly the most likely explanation, but the authors don't actually measure this until Fig. 6E. Perhaps rephrase or make this conclusion later?

Rephrased to: These differences indicate that the slow growth phenotype of the Δ TPC-a mutant may result from a replication defect due to deficient production of essential metabolites like isoprenoids, which are synthesized by apicoplast enzymes. Lines 256-259.

20. Figure 6C and legend. What is the " Δ TPC-cm" condition described in the figure? This isn't described elsewhere. Presumably the Δ TPC-a-iTPC strain mentioned in the legend? The authors should make this clearer.

Corrected.

21. Figure 7B. Figure 7D shows the effects of ionomycin addition on apicoplast [Ca²⁺],

but the equivalent data are not shown for cytosolic $[Ca^{2+}]$. The authors mention that the traces were obtained simultaneously, so why not show this effect?

We did not show the cytosolic Ca^{2+} increase after 1 μM ionomycin because of dye saturation at the concentration used. We performed experiments with a lower concentration (100 nM) of Ionomycin (See figure). We would prefer not to show it in this paper because this is still work in progress. We have found more calcium related phenotypes that we are still investigating.

22. Line 338, “delta symbol” rather than D in “Tati Δ ku80”

Corrected.

23. Figure 8C. The authors should clarify what the arrows depict.

The arrows indicate that the anti-HA EM signal was often found in contact sites.

24. Line 400. “lumen” rather than “matrix”?

Corrected

25. Lines 400-402. “Regardless of the exact mechanism, communication between the apicoplast and the ER is essential for apicoplast function and cell growth and identifies”. As noted in my major comment 4 above, alternatively, it is the compensatory loss of connection between the ER and apicoplast that enables ΔTPC parasites to proliferate. Without data that the observed ER/apicoplast connections are due to loss of TPC and not compensatory mutations that have occurred in this strain, the authors should temper their conclusions.

We now include the ER/Apicoplast connection data of the complemented cells. The data is shown in Figure 8 (see response above)

26. Line 487. Why “plant and animal” whole genome re-sequencing?

That is the service offered by BGI (<https://www.bgi.com/global/sequencing-services/plant-animal-microbial-dna-sequencing/plant-animal-and-microbe-whole-genome-resequencing/>). It provides whole genome service for the organisms (plant and animal) that have reference genomes available. The sentence was corrected.

27. Line 574. In the description of the TEM studies, were the authors blinded to whether

they were observing WT or Δ TPC parasites?

Yes. This is now stated in line 596.

Reviewer #3 (Remarks to the Author):

The revised manuscript has addressed all of the points raised in my previous review.

Andrew Thomas

We thank Dr Thomas for his comments.

Peer Review, third round –

Reviewer #2 (Remarks to the Author):

The authors have undertaken experiments that demonstrate that complementing TPC mutant parasites with a constitutive copy of TPC restores the increase in apicoplast [Ca²⁺] following thapsigargin treatment (Fig 7e-h), and restores apicoplast-ER associations (Fig 8a-b). These data lend support to the hypothesis that TPC (and not compensatory effects of TPC knockout) mediates these processes. This addresses my major concern from the last review round. I have no further comments on the manuscript, and I congratulate the authors on completing an interesting study.